

# EXCESS workshop:
# Descriptions of rising low-energy spectra

**Editors:** A. Fuss[1,2⋆], M. Kaznacheeva[3‡], F. Reindl[1,2†] and F. Wagner[1∘]

⋆ alexander.fuss@oeaw.ac.at, † florian.reindl@oeaw.ac.at,
‡ margarita.kaznacheeva@tum.de, ∘ felix.wagner@oeaw.ac.at

P. Adari[4], A. Aguilar-Arevalo[5], D. Amidei[6], G. Angloher[7], E. Armengaud[8], C. Augier[9],
L. Balogh[10], S. Banik[1,2], D. Baxter[11], C. Beaufort[12], G. Beaulieu[9], V. Belov[13],
Y. Ben Gal[14], G. Benato[15], A. Benoît[16], A. Bento[7,17], L. Bergé[18], A. Bertolini[7],
R. Bhattacharyya[19], J. Billard[9], I.M. Bloch[14], A. Botti[20], R. Breier[21], G. Bres[16],
J.-L. Bret[16], A. Broniatowski[18], A. Brossard[22], C. Bucci[15], R. Bunker[23], M. Cababie[11,20],
M. Calvo[16], P. Camus[16], G. Cancelo[11], L. Canonica[7], F. Cappella[24], L. Cardani[24],
J.-F. Caron[10], N. Casali[24], G.delCastello[24,25], A. Cazes[9], R. Cerulli[26,27],
B.A. Cervantes Vergara[5], D. Chaize[9], M. Chapellier[18,22], L. Chaplinsky[28,29], F. Charlieux[9],
M. Chaudhuri[30], A.E. Chavarria[31], G. Chemin[12], R. Chen[32], H. Chen[33], F. Chierchie[11,34],
I. Colantoni[24,35], J. Colas[9], J. Cooley[36], J.-M. Coquillat[22], E.C. Corcoran[37],
S. Crawford[22], M. Crisler[11], A. Cruciani[24], P. Cushman[38], A. D'Addabbo[15], J.C. D'Olivo[5],
A. Dastgheibi-Fard[12], M. De Jésus[9], Y. Deng[39], J.B. Dent[40], E.L. Depaoli[41], K. Dering[22],
S. Dharani[42,62], S. Di Lorenzo[15], A. Drlica-Wagner[11,43,44], L. Dumoulin[18], D. Durnford[39],
B. Dutta[19], L. Einfalt[1,2], A. Erb[3,45], A. Erhart[3], R. Essig[4], J. Estrada[11], E. Etzion[14],
O. Exshaw[16], F. Favela-Perez[5], F. v. Feilitzsch[3], G. Fernandez Moroni[11],
N. Ferreiro Iachellini[7], S. Ferriol[9], S. Fichtinger[22], E. Figueroa-Feliciano[32],
J.-B. Filippini[9], D. Filosofov[13], J. A. Formaggio[46], M. Friedl[1], S. Fuard[47], D. Fuchs[7],
A. Fuss[1,2], R. Gaïor[48], A. Garai[7], C. Garrah[39], J. Gascon[9], G. Gerbier[22], M. Ghaith[49],
V.M. Ghete[1], D. Gift[4,50], I. Giomataris[8], G. Giroux[22], A. Giuliani[18], P. Gorel[51,52,53],
P. Gorla[15], C. Goupy[8], J. Goupy[16], C. Goy[12], M. Gros[8], P. Gros[22], Y. Guardincerri[11,†],
C. Guerin[9], V. Guidi[54,55], O. Guillaudin[12], S. Gupta[1], E. Guy[9], P. Harrington[46], D. Hauff[7],
S. T. Heine[46], S. A. Hertel[29], S.E. Holland[33], Z. Hong[56], E.W. Hoppe[23], T.W. Hossbach[23],
J.-C. Ianigro[9], V. Iyer[30], A. Jastram[19], M. Ješkovský[21], Y. Jin[57], J. Jochum[58],
J. P. Johnston[46], A. Juillard[9], D. Karaivanov[13], V. Kashyap[30], I. Katsioulas[59],
S. Kazarcev[13], M. Kaznacheeva[3], F. Kelly[37], B. Kilminster[60], A. Kinast[3], L. Klinkenberg[3],
H. Kluck[1], P. Knights[59], Y. Korn[14], H. Kraus[61], B. von Krosigk[42,62], A. Kubik[52],
N.A. Kurinsky[63], J. Lamblin[12], A. Langenkämper[3], S. Langrock[51,52,53], T. Lasserre[8,64],
H. Lattaud[9], P. Lautridou[65], I. Lawson[52], S.J. Lee[60], M. Lee[19], A. Letessier-Selvon[48],
D. Lhuillier[8], M. Li[46], Y.-T. Lin[19], A. Lubashevskiy[13], R. Mahapatra[19], S. Maludze[19],
M. Mancuso[7], I. Manthos[59], L. Marini[15], S. Marnieros[18], R.D. Martin[22], A. Matalon[48,66],
J. Matthews[59], B. Mauri[8], D. W. Mayer[46], A. Mazzolari[55], E. Mazzucato[8],
H. Meyer zu Theenhausen[42,62], E. Michielin[67,68], J. Minet[16], N. Mirabolfathi[19],
K. v. Mirbach[3], D. Misiak[9], P. Mitra[31], J.-L. Mocellin[16], B. Mohanty[30], V. Mokina[1],
J.-P. Mols[8], A. Monfardini[16], F. Mounier[9], S. Munagavalasa[66], J.-F. Muraz[12], X.-F. Navick[8],
T. Neep[59], H. Neog[38], H. Neyrial[8], K. Nikolopoulos[59], A. Nilima[7], C. Nones[8], V. Novati[32],
P. O'Brien[39], L. Oberauer[3], E. Olivieri[18], M. Olmi[15], A. Onillon[8,‡], C. Oriol[18], A. Orly[14],
J.L. Orrell[23], T. Ortmann[3], C.T. Overman[23], C. Pagliarone[15,69], V. Palušová[21], P. Pari[70],
P. K. Patel[29], L. Pattavina[3,71], F. Petricca[7], A. Piers[31], H. D. Pinckney[29], M.-C. Piro[39],
M. Platt[19], D. Poda[18], D. Ponomarev[13], W. Potzel[3], P. Povinec[21], F. Pröbst[7],
P. Privitera[48,66], F. Pucci[7], K. Ramanathan[66,72], J.-S. Real[12], T. Redon[18], F. Reindl[1,2],

---

†Deceased January 2017
‡Now at 3

R. Ren[32], A. Robert[47], J. Da Rocha[48], D. Rodrigues[11,20], R. Rogly[8], J. Rothe[3], N. Rowe[22],
S. Rozov[13], I. Rozova[13], T. Saab[73], N. Saffold[11], T. Salagnac[9], J. Sander[74], V. Sanglard[9],
D. Santos[12], Y. Sarkis[5], V. Savu[8], G. Savvidis[22], I. Savvidis[75], S. Schönert[3], K. Schäffner[7],
N. Schermer[3], J. Schieck[1,2], B. Schmidt[32], D. Schmiedmayer[1,2], C. Schwertner[1,2],
L. Scola[8], M. Settimo[65], Ye. Shevchik[13], V. Sibille[46], I. Sidelnik[76], A. Singal[50], R. Smida[66],
M. Sofo Haro[11,77], T. Soldner[47], J. Stachurska[46], M. Stahlberg[7], L. Stefanazzi[11],
L. Stodolsky[7], C. Strandhagen[58], R. Strauss[3], A. Stutz[12], R. Thomas[66], A. Thompson[19],
J. Tiffenberg[11], C. Tomei[24], M. Traina[48], S. Uemura[11,14], I. Usherov[58], L. Vagneron[9],
W. Van De Pontseele[46], F.A. Vazquez de Sola Fernandez[65], M. Vidal[22], M. Vignati[24,25],
A.L. Virto[78], M. Vivier[8], T. Volansky[14], V. Wagner[3], F. Wagner[1], J. Walker[40], R. Ward[59],
S.L. Watkins[79], A. Wex[3], M. Willers[3], M.J. Wilson[62], L. Winslow[46], E. Yakushev[13],
T.-T. Yu[80], M. Zampaolo[12], A. Zaytsev[42,62], V. Zema[7], D. Zinatulina[13], A. Zolotarova[8]

**1** Institut für Hochenergiephysik der Österreichischen Akademie der Wissenschaften,
1050 Wien, Austria
**2** Atominstitut, Technische Universität Wien, 1020 Wien, Austria
**3** Physik-Department and ORIGINS Excellence Cluster, Technische Universität München,
D-85747 Garching, Germany
**4** C.N. Yang Institute for Theoretical Physics, Stony Brook University, Stony Brook,
NY 11794, USA
**5** Universidad Nacional Autónoma de México, Mexico City, Mexico
**6** Department of Physics, University of Michigan, Ann Arbor, Michigan, USA
**7** Max-Planck-Institut für Physik, D-80805 München, Germany
**8** IRFU, CEA, Université Paris-Saclay, F-91191 Gif-sur-Yvette, France
**9** Univ Lyon, Université Lyon 1, CNRS/IN2P3, IP2I-Lyon, F-69622, Villeurbanne, France
**10** Department of Mechanical and Materials Engineering, Queen's University, Kingston,
Ontario K7L 3N6, Canada
**11** Fermi National Accelerator Laboratory, Batavia, Illinois, USA
**12** Univ. Grenoble Alpes, CNRS, Grenoble INP, LPSC-IN2P3, Grenoble, France 38000
**13** Department of Nuclear Spectroscopy and Radiochemistry,
Laboratory of Nuclear Problems, JINR, Dubna, Moscow Region, Russia 141980
**14** School of Physics and Astronomy, Tel-Aviv University, Tel-Aviv 69978, Israel
**15** INFN, Laboratori Nazionali del Gran Sasso, I-67100 Assergi, Italy
**16** Univ. Grenoble Alpes, CNRS, Grenoble INP, Institut Néel, 38000 , Grenoble, France 38000
**17** Departamento de Fisica, Universidade de Coimbra, P3004 516 Coimbra, Portugal
**18** Université Paris-Saclay, CNRS/IN2P3, IJCLab, 91405 Orsay, France
**19** Department of Physics and Astronomy, and the Mitchell Institute for Fundamental Physics
and Astronomy, Texas A&M University, College Station, TX 77843, USA
**20** Department of Physics, FCEN, University of Buenos Aires and IFIBA,
CONICET, Buenos Aires, Argentina
**21** Comenius University, Faculty of Mathematics, Physics and Informatics,
84248 Bratislava, Slovakia
**22** Department of Physics, Engineering Physics & Astronomy, Queen's University,
Kingston, Ontario K7L 3N6, Canada
**23** Pacific Northwest National Laboratory, Richland, WA 99352, USA
**24** Istituto Nazionale di Fisica Nucleare – Sezione di Roma, Roma I-00185, Italy
**25** Dipartimento di Fisica, Sapienza Università di Roma, Roma I-00185, Italy
**26** Istituto Nazionale di Fisica Nucleare – Sezione di Roma "Tor Vergata", Roma I-00133, Italy
**27** Dipartimento di Fisica, Università di Roma "Tor Vergata", Roma I-00133, Italy
**28** Amherst Center for Fundamental Interactions, University of Massachusetts, Amherst,
MA 01003-9337, USA

**29** Department of Physics, University of Massachusetts, Amherst, MA 01003-9337, USA
**30** National Institute of Science Education and Research (NISER), Jatni 752050, India
**31** Center for Experimental Nuclear Physics and Astrophysics, University of Washington,
Seattle, Washington, USA
**32** Department of Physics & Astronomy, Northwestern University, Evanston,
IL 60208-3112, USA
**33** Lawrence Berkeley National Laboratory,
One Cyclotron Road, Berkeley, California 94720, USA
**34** Instituto de Inv. en Ing. Eléctrica "Alfredo Desages" (IIIE), Dpto. de Ing. Eléctrica y de
Computadoras. CONICET and Universidad Nacional del Sur (UNS), Bahía Blanca, Argentina
**35** Consiglio Nazionale delle Ricerche, Istituto di Nanotecnologia, Roma I-00185, Italy
**36** Department of Physics, Southern Methodist University, Dallas, TX 75275, USA
**37** Chemistry & Chemical Engineering Department, Royal Military College of Canada,
Kingston, Ontario K7K 7B4, Canada
**38** School of Physics & Astronomy, University of Minnesota, Minneapolis, MN 55455, USA
**39** Department of Physics, University of Alberta, Edmonton, Alberta, T6G 2R3, Canada
**40** Department of Physics, Sam Houston State University, Huntsville, TX 77341, USA
**41** CNEA - Gerencia de Área Aplicaciones de la Tecnología Nuclear (GAATN) - Gerencia
Química Nuclear y Ciencias de la Salud - Dpto. Metrologia de Radioisotopos - Division
Metrologia Cientifica Centro Atómico Ezeiza
**42** Institut für Experimentalphysik, Universität Hamburg, 22761 Hamburg, Germany
**43** Kavli Institute for Cosmological Physics, University of Chicago, Chicago, IL 60637, USA
**44** Department of Astronomy and Astrophysics, University of Chicago,
Chicago IL 60637, USA
**45** Walther-Meißner-Institut für Tieftemperaturforschung, D-85748 Garching, Germany
**46** Laboratory for Nuclear Science, Massachusetts Institute of Technology,
Cambridge, MA, USA 02139
**47** Institut Laue Langevin, Grenoble, France 38042
**48** Laboratoire de Physique Nucléaire et des Hautes Énergies (LPNHE), Sorbonne Université,
Université de Paris, CNRS-IN2P3, Paris, France
**49** Department of Physics, Queen's University, Kingston, ON K7L 3N6, Canada
**50** Department of Physics and Astronomy, Stony Brook University, Stony Brook,
NY 11794, USA
**51** Department of Physics and Astronomy, Laurentian University,
Sudbury, Ontario, P3E 2C6, Canada
**52** Sudbury Neutrino Observatory Laboratory (SNOLAB), Lively, Ontario, Canada
**53** Arthur B. McDonald Canadian Astroparticle Physics Research Institute,
Queen's University, Kingston, ON, K7L 3N6, Canada
**54** Dipartimento di Fisica, Università di Ferrara, I-44122 Ferrara, Italy
**55** Istituto Nazionale di Fisica Nucleare – Sezione di Ferrara, I-44122 Ferrara, Italy
**56** Department of Physics, University of Toronto, Toronto, ON M5S 1A7, Canada
**57** C2N, CNRS, Univ. Paris-Sud, Univ. Paris-Saclay, Palaiseau, France 91120
**58** Eberhard-Karls-Universität Tübingen, D-72076 Tübingen, Germany
**59** School of Physics and Astronomy, University of Birmingham,
Birmingham B15 2TT, United Kingdom
**60** Universität Zürich Physik Institut, Zurich, Switzerland
**61** Department of Physics, University of Oxford, Oxford OX1 3RH, United Kingdom
**62** Institute for Astroparticle Physics (IAP), Karlsruhe Institute of Technology (KIT),
76344, Germany
**63** SLAC National Accelerator Laboratory/Kavli Institute for Particle Astrophysics and

Cosmology, Menlo Park, CA 94025, USA

**64** APC, Université de Paris, CNRS, Astroparticule et Cosmologie, Paris F-75013, France

**65** SUBATECH, Université de Nantes, IMT Atlantique, CNRS-IN2P3, Nantes, France

**66** Kavli Institute for Cosmological Physics and The Enrico Fermi Institute,
The University of Chicago, Chicago, Illinois, USA

**67** Department of Physics & Astronomy, University of British Columbia,
Vancouver, BC V6T 1Z1, Canada

**68** TRIUMF, Vancouver, BC V6T 2A3, Canada

**69** Dipartimento di Ingegneria Civile e Meccanica, Universitá degli Studi di Cassino e del
Lazio Meridionale, I-03043 Cassino, Italy

**70** IRAMIS, CEA, Université Paris-Saclay, F-91191 Gif-sur-Yvette, France

**71** GSSI-Gran Sasso Science Institute, I-67100 L'Aquila, Italy

**72** Division of Physics, Mathematics & Astronomy, California Institute of Technology,
Pasadena, California, USA

**73** Department of Physics, University of Florida, Gainesville, FL 32611, USA

**74** Department of Physics, University of South Dakota, Vermillion, SD 57069, USA

**75** Aristotle University of Thessaloniki, Thessaloniki 54124, Greece

**76** CONICET y CNEA - Comisión Nacional de Energía Atómica, Departamento de física de
neutrones, Centro Atómico Bariloche, Av. Bustillo 9500, Bariloche, Argentina

**77** Centro Atómico Bariloche, CNEA/CONICET/IB, Bariloche, Argentina

**78** Instituto de Física de Cantabria (IFCA), CSIC–Universidad de Cantabria, Santander, Spain

**79** Department of Physics, University of California, Berkeley, CA 94720, USA

**80** Department of Physics and Institute for Fundamental Science, University of Oregon,
Eugene, Oregon 97403, USA

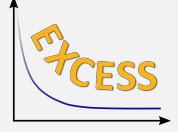

## Abstract

**Many low-threshold experiments observe sharply rising event rates of yet unknown origins below a few hundred eV, and larger than expected from known backgrounds. Due to the significant impact of this excess on the dark matter or neutrino sensitivity of these experiments, a collective effort has been started to share the knowledge about the individual observations. For this, the EXCESS Workshop was initiated. In its first iteration in June 2021, ten rare event search collaborations contributed to this initiative via talks and discussions. The contributing collaborations were CONNIE, CRESST, DAMIC, EDELWEISS, MINER, NEWS-G, NUCLEUS, RICOCHET, SENSEI and SuperCDMS. They presented data about their observed energy spectra and known backgrounds together with details about the respective measurements. In this paper, we summarize the presented information and give a comprehensive overview of the similarities and differences between the distinct measurements. The provided data is furthermore publicly available on the workshop's data repository together with a plotting tool for visualization.**

Received 21-03-2022
Accepted 06-07-2022
Published 09-08-2022

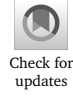
Check for
updates

doi:10.21468/SciPostPhysProc.9.001

# Contents

# 1 Introduction

Modern rare event search experiments have reached sub-keV recoil energy thresholds. As a result, many experiments are observing previously undetected excesses of low-energy events of unknown origin, which clearly surpasses known background levels. Typically starting at energies below 1 keV, the obtained energy spectra rise sharply towards the detector thresholds. Excesses are observed in various experiments with different detector materials, sensors and holding structures, below and above ground, at different temperatures, and background levels. Due to variations in the shape and rate of the excess signal among experiments, detectors and measurements, a dark matter (DM) explanation seems unlikely. Furthermore, the differences in their detailed characteristics point towards multiple origins of the excesses, possibly overlapping between pairs of experiments, but very unlikely in all of them.

An understanding of the observed excesses has a top priority for current low-threshold experiments, since the affected energy region is a crucial part of the region of interest in the search for low-mass DM and detection of coherent elastic neutrino-nucleus scattering (CEvNS). The EXCESS workshop was organized for this purpose. The workshop brought together several experimental collaborations as a joint effort to understand and characterize the observed excesses. It took place $15^{th}$-$16^{th}$ of June 2021 and consisted of 11 talks and 3 discussion sessions. Contributions were provided by the CONNIE [1], CRESST [2], DAMIC [3], EDEL-WEISS [4,5], MINER [6], NEWS-G [7], NUCLEUS [8], RICOCHET [9], SENSEI [10] and SuperCDMS [11–13] collaborations. This original selection of experiments includes those where a common origin of the excess seemed at least plausible based on rate, energy scale and to a lesser extent technology, and those that might be expected to be sensitive to such an excess. However, as no single common origin could (yet) be identified, the EXCESS workshop might benefit from broadening its scope to further experiments. In particular, photon-mediated detectors featuring energy thresholds below the X-ray shell energies (typically sub-keV scale) are of interest, e.g. XENON1T [14].

In the course of the workshop, collaborations agreed to share the data of their most recent observations. The data is collected in a GitHub repository, together with tools for the collective visualization [15]. The data was shared under the CC BY 4.0 license, allowing for its use in independent publications, as long as referenced properly. All presentations are found on the workshop's Indico webpage [16].

In section 2, we review the specifics of the individual measurements, whose data are shown and compared in section 3. We will finish this report by giving an outlook on the further activities of the EXCESS workshop in section 4.

**Related work.** These low energy excesses have recently sparked a lot of interest in the community and have been the topic of several independent publications. In Ref. [17], the authors explore the possibility of a dark matter origin through a plasmon scattering channels, which has since been ruled out by Refs. [18,19]. In Ref. [20], they revise this interpretation, and disfavor any common nuclear recoil origin in semiconductor targets, based on new data and analysis methods. In parallel, Ref. [21] proposes a test of the origin of the recoil, based on material-dependent energy loss of nuclear recoils due to crystal defects. While mentioned work has focused on the search for potential origins of the excess, the subject of our work is the detailed description of the corresponding low energy spectra. By this, we seek to set a better foundation for the search for potential origins.

# 2 Experimental observation of rising low-energy spectra

In the following, we report several experimental observations of low-energy spectra featuring potential excesses above known background levels. The presented results are obtained in different environments and with different detector concepts. Thus, we introduce some global terminology to facilitate the understanding and interpretation of the subsequent subsections.

A general way to categorize the various experiments is to describe the location at which measurements are performed: *above ground* and *below ground*. The latter can be further distinguished between *shallow* and *deep underground* sites. The depth of the location is usually given in meter water equivalent (m.w.e.) and has an impact on the overall environmental background level, especially with respect to cosmic-ray induced particles. While DM experiments are mostly located deep underground, CE$\nu$NS experiments usually take data above ground. Additionally, prototype measurements for both of these rare event searches are often performed in above ground or shallow underground facilities.

The energy scales measured, as well as the type of energy deposition, provide another important distinction between experiments. Depending on the type of signal (i.e. heat, charge, light or a combination of any of these), type of calibration and capability for event-by-event particle discrimination, an energy deposition may be measured in units of *total energy, nuclear recoil equivalent energy, or electron equivalent energy*. As nuclear recoil signals measured via charge or light are quenched with respect to electron recoils, this can have an impact on the interpretation of and comparability between results of different measurements. While the parameters for the conversion between energy scales are well studied for most materials, applying the conversion is always based on assumptions regarding the origin of the measured signals. Thus, an unbiased comparison between experiments measuring different energy scales is difficult.

The so-called *detector efficiency* is another factor that is taken into account, to the best of the knowledge of each experiment, to allow for comparison between different experimental data. This generally energy-dependent parameter describes the probability for an event in the detector to appear in the final spectrum shown. Spectra provided by the experiments are corrected for this factor to represent the actual rate in the detectors. However, the type and exact definition of cuts may differ between the experiments and could have an impact on comparing different results close to the thresholds.

In the following, the various detector concepts (cryogenic, CCD, or gaseous SPC detectors) and the individual experimental observations that were presented in the EXCESS workshop are described.

## 2.1 Cryogenic Detectors

Cryogenic detectors measure the deposited energy via a temperature rise $\Delta T$ in a sensor caused by a particle interaction in the target material. In order to improve the sensitivity, a low heat capacity $C$ and therefore a low operating temperature are required. The detectors are usually formed by a crystal target operated at temperatures below 50 mK. Several methods are available to measure the amount of energy deposited in the target crystal, i.e. to convert the energy released to a readable temperature signal, three of which are described next.

**Neutron-transmutation-doped (NTD)** sensors are thermal sensors made out of a Ge crystal doped by an intense neutron irradiation [22]. This process introduces highly homogeneously distributed impurities in the semiconducting Ge crystal, which leads to a strong temperature dependence of the resistance at cryogenic temperatures. After a particle interaction causes a temperature rise in the target crystal, the resistance of the NTD decreases and the signal is read by the change of its voltage bias [23]. NTD sensors are used in the EDELWEISS and Ricochet experiments described below.

**Transition-Edge Sensors (TES)** offer another type of temperature sensor. TES are typically sensitive enough to register temperature changes of less than 0.1 mK [24]. They consist of a superconducting film stabilized at a temperature that lies within its steep transition from the superconducting to the normal conducting state. In this case, a small temperature rise causes a fast and measurable resistance increase. When deposited on a crystal substrate, the signal is dominated mainly by athermal phonons (i.e. phonons are captured before thermalizing) resulting from some type of particle interaction within the substrate. TESs are used by the CRESST and NUCLEUS experiments described below.

**Quasiparticle-trap-assisted Electrothermal-feedback TES (QET).** To increase the total sensor surface area without increasing the sensor heat capacity, a TES can be fabricated with a small overlap region with superconducting fin structures (typically Al), forming a QET [25]. The superconducting fins collect the athermal phonons, which break cooper pairs to create quasiparticles that diffuse through the fins until reaching the TES and thermalizing. Because the thermal coupling between the superconducting fin and the TES is very poor, the thermal coupling between the TES and the absorber dominates. Thus, baseline energy resolutions on the order of what is expected from the intrinsic TES noise can be reached while operating with larger sensor areas. This detector concept is used in the SuperCDMS-CPD, SuperCDMS-HVeV and MINER DM searches described below.

Growing the target crystals from a scintillating or semiconducting material provides an additional signal channel that can be used for event discrimination.

In case of **scintillating material**, a fraction of the energy deposited in the target by a particle interaction will be released as *light*. The amount of light depends on the type of the recoil: if an incoming particle scatters off an electron the amount of light produced for a given deposited energy is significantly larger than if it scatters off a nucleus, a phenomenon known as quenching. The light produced by particle interactions in scintillating crystals is measured by a separate cryogenic light detector, which enables particle discrimination on an event-by-event basis. This approach is used in the CRESST experiment.

If a **semiconductor material** is used as a target, both *heat* and *charge* are produced in the detector volume by a particle interaction. If no electric field is applied across the crystal, then a recoil with a nucleus produces both phonons and electron-hole pairs, which quickly recombine into phonons as well. These phonons then travel throughout the crystal, downconverting from optical to acoustic phonons and eventually thermalizing in the substrate.

In the presence of electric field applied to electrodes covering the detector, generated electron-hole pairs drift across the crystal providing an ionization signal. Moreover, this drift causes an amplification of the number of phonons that increases the measured heat energy by an additional term $E_{NTL} = N_{eh}V_{bias}$, where $N_{eh}$ is the number of the electron-hole pairs produced and $V_{bias}$ is the voltage bias applied to the electrodes. This effect is known as **Neganov-Trofimov-Luke (NTL) amplification** [26, 27]. In case of $V_{bias} \neq 0$, measured ionization $E_{ion}$ and heat $E_{heat}$ energies can be expressed as:

$$E_{ion} = Y^i(E_R) \cdot E_R \text{ and } E_{heat} = E_R + E_{NTL} = E_R + N_{eh} \cdot e \cdot V_{bias}. \tag{1}$$

The total number of electron-hole pairs created in an interaction is typically determined as per

$$N_{eh} = Y^i(E_R)\frac{E_R}{\epsilon_{eh}(E_R)}, \tag{2}$$

where $\epsilon_{eh}$ is the average energy required to produce one e-h pair and $Y^i$ is the ionization yield. The value of the ionization yield $Y$ depends on the nature of the recoil $i$: $Y^i(E_R) = Y = 1$ for electron recoils and takes significantly smaller values for nuclear recoils.

For both light and ionization approaches, event discrimination has only a limited power for the recoil energies below 1 keV. However, modern technologies, e.g. HEMT preamplification for charge readout, are expected to lower this threshold significantly.

A large share of the measurements described at the EXCESS workshop follow a cryogenic detector concept. We describe them in the following subsections 2.1.1-2.1.6.

### 2.1.1 CRESST-III

*Section editor: Christian Strandhagen (christian.strandhagen@uni-tuebingen.de)*

The results shown here pertain to the module *Det-A* operated in the first data taking period of CRESST-III (from 05/2016 until 02/2018). The experimental setup at Laboratori Nazionali del Gran Sasso (LNGS), the CRESST-III detector concept, the data acquisition and analysis of this module are described in much detail in [28].

**Detector concept and setup**  CRESST detectors are operated in a shielded cryostat located in hall A of the LNGS underground laboratory. The rock provides a water-equivalent overburden of 3600 m. In addition to a layered passive shielding consisting of polyethylene, lead and copper, the setup is surrounded by plastic scintillator panels acting as a muon veto. With the exception of a small hole on top accommodating the neck of the cryostat, this covers the entire part where the detectors are hosted (98.7 % geometric coverage) [29, 30].

A CRESST-III module is made up of two individual cryogenic particle detectors: the main absorber - also called *phonon detector* - made out of a scintillating crystal (in this case $CaWO_4$) with a dimension of (20 x 20 x 10) mm$^3$ and a silicon-on-sapphire *light detector* which covers one face of the absorber crystal but is much thinner ((20 x 20 x 0.4) mm$^3$). Both phonon and light detector are equipped with a tungsten TES which is directly evaporated on the material and is operated at  15 mK and read out using SQUIDs. Alongside the TES, there is also a heater which is necessary to stabilize the detector in its operating point in the superconducting transition. This is done by periodically sending large voltage pulses to the heater which drive the TES out of transition and adjusting the heater power such that the pulse height of these control pulses remains constant. In addition to these large control pulses also smaller so-called test pulses with known energy are injected to the heater in regular intervals which allow to linearize the energy scale of the detector and to correct small fluctuations of the detector response over time.

The two detectors are held by $CaWO_4$ sticks with a diameter of 2.5 mm and a rounded tip, which are pressed onto the detectors with bronze clamps from the outside through holes in the copper housing completely surrounding the entire module. The inside of this housing is covered with a reflective and scintillating foil (Vikuiti by 3M) which enhances the light collection and allows to discriminate surface alpha background. For the module discussed here, also each stick holding the main absorber crystal is equipped with a small TES and is operated as a cryogenic detector itself. This opens up the possibility to identify and veto events which occur in the sticks that could induce a smaller signal in the main absorber due to transmission of phonons via the interface. A schematic drawing of the detector module is shown in Fig. 1.

**Data acquisition and processing**  In CRESST-III, the data for all modules are continuously sampled with a frequency of 25 kHz and triggered offline with an optimum filter which takes into account the signal pulse shape and the noise power spectrum of each detector. The thresholds were adjusted such that one noise trigger per kg day exposure is expected [31]. The optimum filter amplitude is then directly used to determine the energy of the events in the low energy region where the detector response is roughly linear. For higher energies, where pulses become saturated due to the nature of the transition curve, a truncated template fit was

---

[1]Reprinted figure with permission from [28]. Copyright 2019 by the American Physical Society.

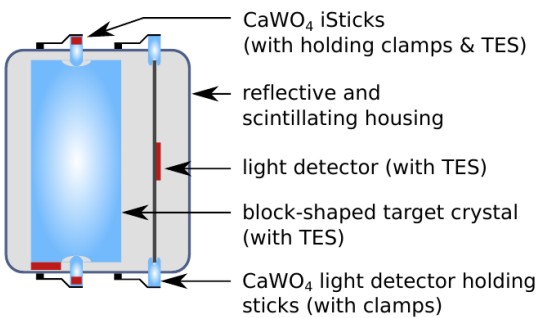

Figure 1: Schematic side view of Det-A operated in CRESST-III. Figure from Ref. [28][1].

developed in the past to better reconstruct the amplitude of high energy events [32]. Both energy scales are first calibrated using the heater test pulses and are then matched to each other.

Finally, the energy scale has to be converted from the energy input from the heater to the deposited energy of particle events. For this, a $^{57}$Co source located outside the shielding providing a gamma line at 122 keV and a tungsten escape peak at 63.2 keV was used. In the case of Det-A the energy scale was fine-adjusted using the 11.27 keV peak which originates from the cosmogenic activation of tungsten [33]. A unique feature of CRESST is that the energy scale for nuclear recoils and electron or gamma events is practically the same. The maximum difference of the energy scales is given by the light output of the crystal, which is typically around 5 % for CaWO$_4$ crystals.

Light detectors usually are not calibrated in absolute energy but in *electron equivalent* energy (denoted as keV$_{ee}$), which is the total energy detected in the phonon detector corresponding to the energy detected in the light detector in form of light from a electron or gamma event of a given energy. This is then used to calculate the *light yield* which is obtained by dividing the light energy (in keV$_{ee}$) by the energy measured in the phonon detector. This quantity is then approximately one by definition for electron/gamma events from the calibration source and smaller for nuclear recoil events because of the reduced light output due to quenching.

For the data selection, first, periods where the detector was not operated in stable conditions are discarded. This can be periods with known external disturbances, with exceptionally high noise or where the detector was not in the correct operating point. Then some quality cuts are made to ensure that only valid pulses where the energy can reliably be reconstructed are selected. These cuts mainly remove artifacts introduced by the electronics or pile-up events. Finally, events coincident either with the muon veto or with other cryogenic detectors (including the instrumented holding sticks) are discarded. To account for the possibility of removing also potential signal events by these cuts, the survival probability of signal-like events is determined by superimposing signal templates scaled to different amplitudes at random positions onto the data stream. Then the same selection criteria as to the real data are applied the fraction of events surviving is calculated as a function of energy.

Since the trigger is also done offline in software it is also included in this simulation procedure. The quoted energy threshold of the detector is defined as the (simulated) energy where 50 % of the injected pulses are triggered. This definition takes into account small gain variations which are accounted for in the analysis and the simulation procedure.

To have an in-situ measurement of the nuclear recoil bands another calibration with a neutron source is performed in each run. The pulse shape of the neutron induced nuclear recoil events is the same as for electron/gamma events, so all cuts should affect both event

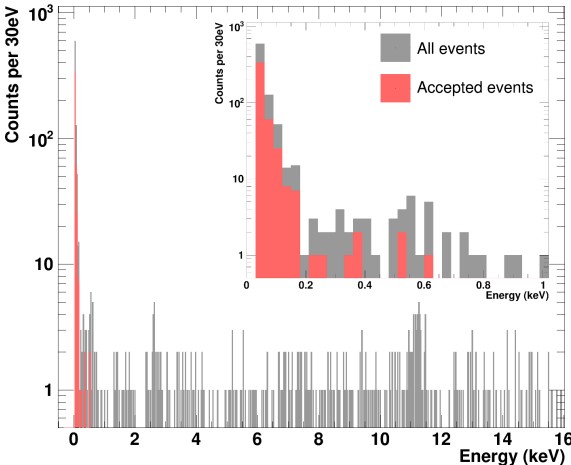

Figure 2: Energy spectrum of the DM data set of Det-A of CRESST-III. Shown in gray are all events and in red only the events in the acceptance region for the DM analysis Figure from Ref. [28][2].

classes in the same way.

**Energy spectrum from Det-A**  In the workshop, data from the module Det-A operated in the first data taking period of CRESST-III were presented. The module is based on a $CaWO_4$ crystal with a mass of 23.6 g and achieved a baseline energy resolution of $\sigma = 4.6$ eV [28]. With this the offline trigger threshold was set to a value corresponding to an energy of 30.1 eV for nuclear recoils using the method outlined in [31]. The exposure before cuts used for the DM analysis with Det-A amounts to 5.594 kg·day. The average survival probability for signal events (neglecting the energy dependence introduced by the trigger efficiency) is ∼65 %. The data of this module in the energy range from 30.1 eV up to 16 keV are published at [34].

Fig. 2 shows the energy spectrum of all events in the DM data set of Det-A from [28]. At energies below 200 eV one can observe a sharp rise of the event rate. The average pulse shape of these events can not be distinguished from particle-induced events at higher energies. Due to the low light output at these energies it is impossible to tell if these events are caused by nuclear recoils or by electron/gamma events. According to the noise model a total of 3.6 events from noise triggers are expected, which is much less than what is observed. A similar event population is observed in all other detectors operated in the same run which had a sufficiently low energy threshold. However, the rate and spectral shape of the excess contributions is not compatible between these different detectors which disfavours a common origin of these events (like e.g. a DM signal) [32, 35].

Various hypotheses for the origin of these excess events have been put forward and are currently explored with specifically modified detector modules. Among the sources that are discussed are effects related to stress in the crystal lattice, stress induced by the holders, scintillation light produced in the vicinity of the absorber crystal (but not detected by the light detector) and low energetic surface background. To investigate these, absorber crystals with different materials or grown under different conditions are used, the holding scheme is modified and modules without any scintillating materials are employed. In addition there are ongoing studies of the time-dependence of the excess rate.

---

[2]Reprinted figure with permission from [28]. Copyright 2019 by the American Physical Society.

### 2.1.2 EDELWEISS and Ricochet-CryoCube

*Section editors: Julien Billard (j.billard@ipnl.in2p3.fr), Jules Gascon (j.gascon@ipnl.in2p3.fr)*

This section describes the experimental setup and the data collection used by the EDEL-WEISS collaboration for its above-ground searches with the detector RED20 [4] and its underground searches for interactions with electrons with the detector RED30 [5]. The following discussion and observations also apply to the CryoCube detector array [36] of the future Ricochet experiment [37] using similar HPGe cryogenic detector tuned for above-ground operations in the context of CE$\nu$NS searches at nuclear reactors.

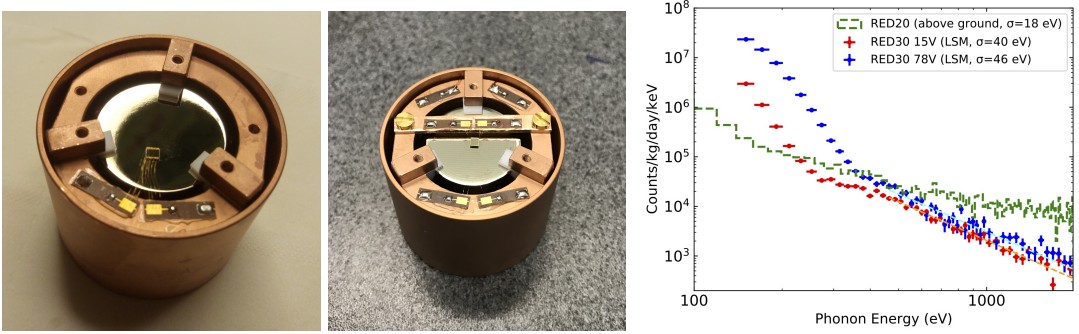

Figure 3: **Left** and **center** panels: picture of the 33.4 g EDELWEISS detectors RED20 and RED30, respectively, in their copper holder. The exposed side shows the Ge-NTD thermistances glued on the top side. **Right**: Efficiency-corrected event rates, in events per kgd and per keV, as a function of the total phonon energy in eV. Green dashed: RED20. Red and blue points: spectra recorded by RED30 with biases of 15 and 78 Volt, respectively. The rises below 300 eV correspond to the onset of the read-out noise. The lines are fits to guide the eyes.

**Detector concept and setup** The absorbers of both detectors are 33.4 g ultra-pure germanium cylindrical crystals with a diameter of 20 mm and a height of 20 mm. The thermal sensor is a Ge-NTD thermistance of $2 \times 2 \times 0.5$ mm$^3$, glued on the top surface of the crystal. The electrical contacts are gold wires bonded to the Ge-NTD on one side and to gold pads deposited on a Kapton tape glued to the copper housing of the detector, on the other side. The thermal link between the absorber and the housing goes through the Ge-NTD and these wires. It is dimensioned as to result in a main decay time constant of the order of 20 ms, sufficiently large compared to its risetime of ~6 ms. The crystal of RED20 (see Fig. 3 left panel) is held by six L-shaped PTFE clamps (three on the top and three on the bottom), each having a mass of 50 mg. For RED30 (see Fig. 3 middle panel), the three bottom clamps were replaced with sapphire spheres with a diameter of 3.18 mm, held up by chrysocale clamps. The voltage drop across the Ge-NTD is measured differentially via a biFET cooled-down to ~100 K. To further cancel common electronic noise, the current is modulated from positive to negative values following a square wave function with a frequency chosen as to optimise the baseline resolution. The chosen frequency were 400 and 500 Hz for RED20 and RED30, respectively.

The main difference is that RED30 is equipped with electrodes to enhance the thermal signal using the Neganov-Trofimov-Luke effect [26,27]. Two aluminum electrodes were photolithographed on each of the two planar surfaces: a central electrode in a grid layout (square meshing with a 500 $\mu$m pitch), and a guard electrode made of a concentric ring on the outer edges of the surface. A $2\times 2$ mm$^2$ area was left empty at the center of one face to allow for

the direct gluing of the Ge-NTD on the germanium surface. The grid pattern was chosen as to keep the fraction of the surface covered by electrodes to 4%. Each electrode is biased and read out separately.

Lastly, the Ricochet CryoCube detector prototypes are 42 g Ge cylindrical crystals of 30 mm diameter and 10 mm height. They are mounted in their copper holders using 9 sapphire balls (3.18 mm diameter), 3 on top, 3 on the bottom and 3 on the sides with adjustable clamping force. With such holders, the detectors are found to be insensitive to pulse tube induced vibrations and frictions even without using any cryogenic suspension. However, despite of these improved holding scheme, no improvements on the rate of excess events at the lowest energies was seen [36].

**Data acquisition and processing**    The data acquisition system and readout electronics are described in detail in [38]. The data from the phonon and ionization channels are digitized at a frequency of 100 kHz, filtered, averaged, and continuously stored on disk with a digitization rate matching the Ge-NTD modulation frequency. Events are identified off-line using optimal filters based on the measured noise PSDs and the pulse shape of the heat signal. The events are searched for and selected iteratively using a decreasing energy ordering rule. At each iteration, the data within a given time window of width $\pm \Delta T$ is excluded from further pulse searches. The value of $\Delta T$ depends on the typical rate in the detector: 2 s for the data recorded in the underground laboratory, and 1 s for the above-ground laboratory data. The iterative trigger search procedure stops when a given minimal significance threshold is reached, or if there is no time interval greater than $\Delta T$ left in the stream. For each event, the pulse amplitudes of the active channels (heat and ionisation) are obtained by minimizing the $\chi^2$ in the frequency domain, using the known noise PSD and pulse shapes, and assuming a common pulse starting time in the case of multiple channels. The energy dependence of the trigger, as well as all other biases induced by the data reconstruction and complete analysis procedure, are taken into account by measuring the response for pulses with well-defined energies injected at random times throughout the entire real data streams and subjected to the same triggering, data selection and reconstruction as the real data. The procedure also measures the systematic shift in energy that appears when the signal amplitude approaches that of typical noise fluctuations, see [4] for a more detailed discussion of the processing pipeline. For a reliable bin-to-bin comparison with the other energy spectra in the EXCESS database, the selected ranges are restricted to those where the shift between the true and reconstructed energies are smaller than the energy bins. Event populations with distinctive pulse shapes (such as interactions occurring within the Ge-NTD sensor, by pulses injected through the clamps or glitches in the digitization) were removed using cuts based on the $\chi^2$ obtained using different pulse shape hypotheses [4, 5].

**Energy spectrum from RED20**    The detector RED20 was operated in the dry dilution cryostat of the Institut de Physique des 2 Infinis de Lyon (IP2I) installed in a surface building [4]. The overburden consists of 20 cm (40 cm) of concrete from the ceiling (walls) and 10 cm of lead shielding which surrounds the detector in all directions, apart from an opening of around 50° above the detector. After a period of two weeks devoted to the cool-down and to detector studies, the suspended support structure of the detectors was kept at a regulated temperature of 17 mK for a period of 6 days. The data in a 24-h period (0.033 kg days) near the end was blinded for strongly interacting DM searches. The remaining 5 days were used to tune the analysis procedure, the selection cuts and the energy search intervals. The average baseline energy resolution throughout the 6-day period is 18 eV (*rms*), with a 3% overall decrease from beginning to end. The average value during the blinded data period is 17.7 eV. The energy resolution measured at 5.9 keV with a $^{55}$Fe source is 34 eV (*rms*). In the absence of NTL effect,

this energy scale is directly applicable to both electronic and nuclear recoils. The observed energy spectrum of RED20 is shown as the green histogram on Fig. 3 (right panel). The event rate at 200 eV is $10^5$ count/keV/kg/day, decreasing to $10^4$ count/keV/kg/day at 1 keV. This is consistent with spectra obtained with RICOCHET-CryoCube detector prototypes of similar design operated in the same above-ground cryostat [39]. Additionally, with the particle identification capabilities of the CryoCube detectors, it was found that the observed background level at IP2I is well described by a flat gamma contribution of about 5000 count/keV/kg/day and a rising neutron background of about 1000 count/keV/kg/day at 15 keV and 5000 count/keV/kg/day at 1 keV [40]. Interestingly, the neutron background above 1 keV is observed [39] to be consistent to within a factor of two (assuming all neutrons are of cosmogenic origin) with CRY-based [41] cosmogenic induced neutron background simulations.

**Energy spectrum from RED30**    The detector RED30 [5] has been part of the payload of a 19-month cool-down of the EDELWEISS-III cryostat [38] in the Laboratoire Souterrain de Modane (LSM). The underground site is protected by a water-equivalent overburden of 4800 m. The cryostat is completely covered by a layer of at least 50 cm of polyethylene and 20 cm of lead. Prior to its installation at LSM, the detector was uniformly activated using a neutron AmBe source. This produces a uniform population of $^{71}$Ge throughout the detector volume, which subsequently decays by electron capture in the K, L, and M shells with a half-life of 11 days. The observed de-excitation lines at 10.37, 1.30, and 0.16 keV were used to calibrate the non-linearity of the energy scale and provide an independent cross-check of the efficiency derived from the pulse simulations. More importantly, the dominant 10.37 keV K-shell population provides a clean sample of mono-energetic single-site electron recoils to quantify the charge collection performance as a function of the applied bias, and its evolution in time. These studies were performed in the first five months of the cool-down, while the activation was the most intense. The baseline resolution of the phonon signal is 35 eV ($rms$) at 15 V and 44 eV at 78V. Given the NTL amplification, this corresponded to a $rms$ resolution in the energy scale relevant for electron recoil (eV$_{ee}$ $i.e.$ keV-electron-equivalent) of 1.63 eV$_{ee}$, or 0.54 electron-hole pairs. The resolution at 160 eV$_{ee}$ (M-shell) is 8 eV$_{ee}$, consistent with an expected Fano factor of 0.15. Seven days of data were recorded at 78V, while the temperature was kept at 20.7 mK. 89h of data were selected for the stability of the baseline resolution. It was split in a blind sample of 58h, sandwiched between two non-blind intervals used to optimise the analysis procedure and cuts. The heat resolution in the blind sample is 1.58 eV$_{ee}$ (0.53 electron-hole pair). Three days after the search, the detector was exposed again to a strong AmBe source for 15 h, in order to confirm the stability of the detector response and to provide a sample of 858 reference K-shell events for the pulse simulation. A fraction of 19% of that sample exhibits a degraded charge collection. The contribution of this population to the signal efficiency was set to zero to set conservative DM limits, but should be kept in mind when interpreting the RED30 spectrum in terms of electron recoils. Figure 3 (right panel) shows the comparison of the spectra recorded at different bias voltages of 15 V (red) and 78 V(blue). Also shown as light red and blue dotted lines are the fit to the tails of the two event distributions beyond 500 eV in total phonon energy. As can be concluded from this comparison, the two energy spectra have similar shape when expressed in terms of total phonon energy even though they have NTD gains differing by a factor 4.5. This is a strong indication that most of the events observed above 15 eV$_{ee}$ (400 eV in total energy) are not associated with the creation of electron-hole pairs in the detector. Lastly, as can be concluded from Fig. 3, this non-ionizing low-energy excess from RED30 is a mere factor of 3 lower than the event rate observed above ground with RED20. As a matter of fact, the only benefit from being well shielded within an underground experiment appears for energies beyond 1 keV where a background reduction reaching more than two orders of magnitude was observed.

**Discussion**   The observation of a large population of events with no charge signal had been first documented by EDELWEISS-III with its 860 g detectors [42], above a threshold of 5 keV. At such energies, the absence of ionization could be easily confirmed using the 230 eV$_{ee}$ resolution of the ionization channels. The observed spectra in RED20 and RED30 extend to a much lower threshold, well beyond the reach of the EDELWEISS JFET-based ionization resolution, allowing more in depth studies of this yet-to-be-explained excess. Over the last couple of years, the EDELWEISS and Ricochet collaborations have jointly performed additional studies on numerous detector prototypes and were able to gather some valuable information. Namely, the shape of the spectra does not vary significantly in time, while the absolute rate decreases slowly over time. Sudden increases were observed at times after warming-up the detectors above 10 K. Following such cryogenic events, the rates observed in RED30 was correlated with those observed simultaneously in other, more massive detectors (200g and 860g), but no coincident events between detectors were observed. Various numbers of detector holding strategies have been tested, both within above and underground operations, with adjustable stress on the crystal and with different materials using, or not, cryogenic suspensions. However, none of these tests have shown a significant effect on the magnitude and shape of this low-energy excess. As of today, one of the main hypothesis on its origin, along with others to be tested, is the cracking of the epoxy used to glue the Ge-NTD on the crystal. In parallel to these studies, identification strategies based on improved ionization resolutions [36, 43] and on superconducting single-electron sensors are currently under development.

### 2.1.3   MINER

*Section editor: Rupak Mahapatra (mahapatra@physics.tamu.edu)*

The Mitchell Institute Neutrino Experiment at Reactor (MINER) is a reactor based experiment at Texas A&M university that combines well-demonstrated low-threshold cryogenic phonon-based detector technology developed for the SuperCDMS Dark Matter search experiment with a unique megawatt research reactor that has a movable core providing few meter-scale proximity to the core. The low-threshold detectors will allow detection of coherent scattering of low energy neutrinos that is yet to be detected in any reactor experiment. These high resolution detectors, combined with a movable core, provide the ideal setup to search for short-baseline sterile neutrino oscillation by removing the most common systematic in current experiments, the reactor flux uncertainty. Very short baseline oscillation will be explored as a ratio of rates at various distances, with expected standard model (SM) rates and known scaling of background. Hence MINER will be largely insensitive to absolute reactor flux. Additionally, low variation in a MW research reactor power combined with meter-scale proximity to the core provides much better systematics compared to a GW power reactor, where the typical detector to core distance is of the order of 30 meters or higher resulting in similar neutrino flux incident on a detector. Utilizing multiple targets (Ge/Si/Al$_2$O$_3$) allows for detailed understanding of the signal and backgrounds in the experiment. Precise understanding of the background is important for searches of Non Standard Interactions (NSI) through a small additional signal.

Phase-1 of the MINER experiment is already operational as a demonstration experiment with a 2-kg (4-kg maximum capacity) payload at a distance of approximately 4.5 m from the reactor core, that would provide a signal rate approaching 1000 events per year and a target background of 100-1000 counts/keV/kg/day. Phase-2 of the MINER experiment experiment will have a 20 kg payload (inside a 30-kg infrastructure) that would be housed inside a hermetically shielded ice-box connected through a cold finger to the mixing chamber of the Bluefors fridge. This would provide a proximity of approximately 2 m. The operational 2-kg demonstration phase provides an excellent opportunity to design the full MINER experiment with 10x larger payload, 10x higher flux due to proximity to core and 10x lower background

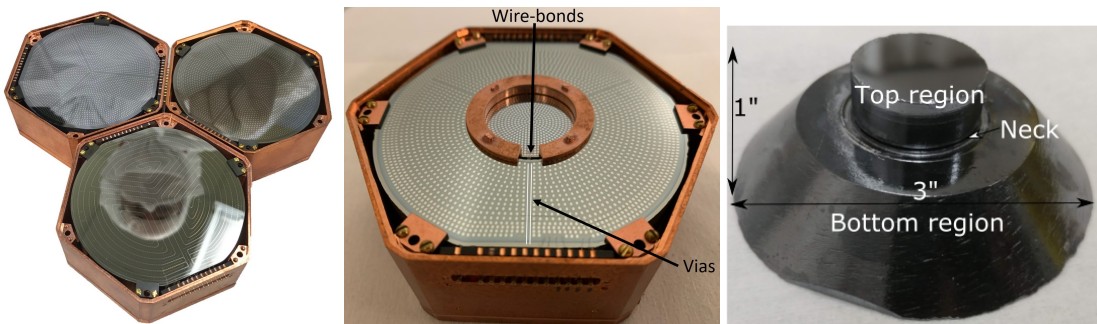

Figure 4: MINER detector technology: **(left)** SuperCDMS detector technologies in use in the form of iZIP and HV detectors, **(center)** New MINER technology that provides significantly improved background reduction, **(right)** and rejection.

due to hermetic passive and active shielding. The sensitivity to CE$\nu$NS will improve by at least two orders of magnitude, allowing for precision tests of eV-scale sterile-$\nu$, Non-Standard Interactions and neutrino magnetic moment.

**Detector concept and setup** The MINER detector technology follows the same principle as is used in the successful SuperCDMS experiment. The MINER detector payload is a combination of many detector technologies all of which use superconducting tungsten TES that operate through QET feedback mechanism using athermal phonons. SuperCDMS High Voltage (HV) Neganov-Luke phonon-assisted ionization detectors provide low threshold (sub 100 eV) and no discrimination. In addition, one iZIP (Z-sensitive ionization + interleaved phonon) detector [44] with electron recoil versus nuclear recoil background discrimination is deployed during each MINER run to monitor the neutron backgrounds down to 1 keV, providing excellent background estimation and validations for simulations inside the signal region of interest (ROI) of 100 eV - 1200 eV (Ge)/3100 eV (Si). A newer generation of sapphire detectors (ROI 10-4100 eV) with expected thresholds of sub-50 eV form the bulk of the detector payload providing strong sensitivity to signal.

To operate and read out the detectors, and to amplify the detector signals, MINER uses re-purposed cold electronics from the decommissioned SuperCDMS Soudan experiment. This includes SQUID-based phonon signal amplifiers and cold FET-based ionization signal amplifiers. Their noise performance has been demonstrated to be better than required for our threshold goals. The data presented in this paper are from the operation of the sapphire detectors at 0V, using phonon sensors only. The measured energies provide the true recoil energies after calibration, without any Lindhard suppression.

Phonon sensors cover the entire surface of each side of a detector and are grouped into four separate channels on each side. Each of the four phonon channels is connected via a cold SQUID-based amplifier to the room temperature electronics. This partition of phonon sensors allows for the event localization in the crystal using either the timing or the relative amplitude of the signals. The outer phonon channel amplitude and timing relative to the three inner channels can be used to identify the events near the fringing faces of the detector and to define the fiducial volume of the detector.

The detectors are calibrated using low-energy gamma sources like $^{55}$Fe and $^{241}$Am sources, as well as external $^{60}$Co and $^{252}$Cf sources. The lowest baseline resolution has been achieved in the approximately 100 g sapphire detectors with as low as 15 eV, thus an expected detection threshold of sub 50-eV. The sapphire detectors do not suffer from any Lindhard suppression and hence the quoted threshold is the recoil energy threshold for CE$\nu$NS processes. Initial studies have been carried out in our test facility to measure the light from the sapphire detectors using

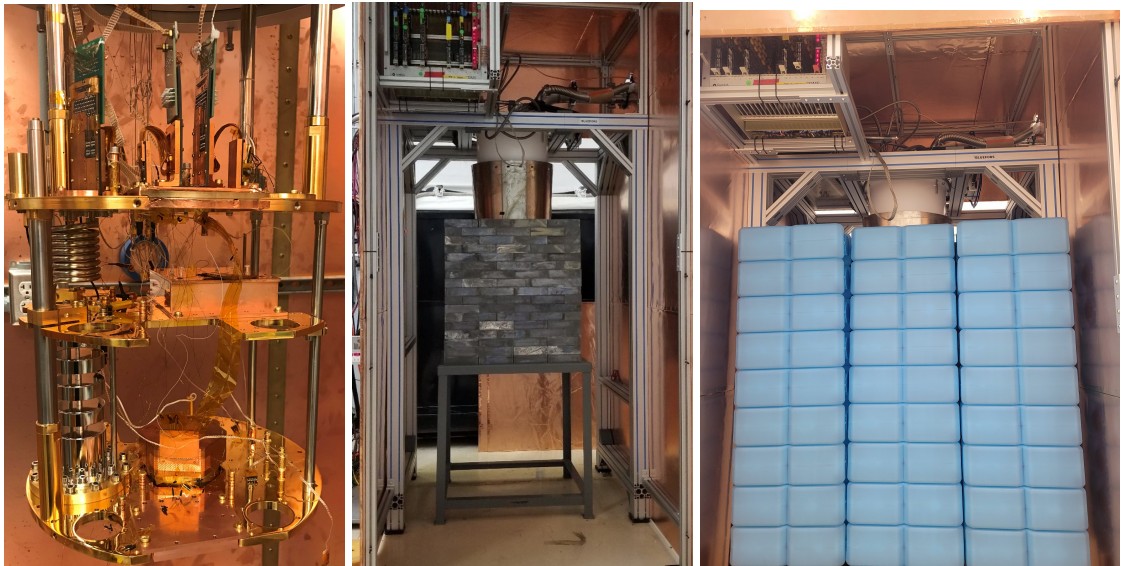

Figure 5: **(left)** Typical detector mounting inside the MINER Bluefors fridge, similar to how SuperCDMS test facilities mount the detectors. The SQUIDs are mounted at the 600 mK stage. The detector stack is designed to provide 1-inch internal hermetic shielding surrounding the detectors to reduce radioactive background, **(center)** 4-inch lead shielding surrounds the fridge with an open top, **(right)** 55% borated rubber sheets of 2mm thickness surround the lead shielding to capture thermal neutrons. Water bricks of 8-inch thickness surround the entire setup except for the open top. External neutrons are moderated by the water shielding and then captured efficiently by the borated rubber shielding, with the lead shielding providing shielding against external gammas and the gammas from the thermal neutron capture in boron.

adjacent Si high voltage (HV) detectors that show good linearity of the light signal with voltage on the Si HV detectors.

**Low energy excess without a donut active veto**  While the best background performance was achieved with the smaller germanium coin detector of approximately 25 g mass, surrounded by a hermetic active germanium veto of 1 inch width, the data was lacking in low-energy performance due to the DAQ trigger threshold. Recent runs have been carried out with an upgraded DAQ system that is capable of running in triggerless mode on a large number of phonon channels. To gain on the fiducial mass and with the restrictions imposed by the maximum mass that can be suspended directly from the Mixing Chamber, it was opted to forgo the coin style detector housed inside a fully hermetic shielding. Instead it was chosen to deploy full sized (100-200 g) sapphire detectors assembled in a typical tower like configuration with a half inch hermetic passive copper shielding. The data analysis uses single scatter events from among the tower of 5 sapphire detectors to study the background at low energies.

The experiment uses thinner ( 100 g) sapphire detectors of approximately 4 mm thickness and thicker ( 250 g) sapphire detectors of approximately 1 cm thickness. The thin detectors have provided as low a resolution as  15 eV, depending on the environmental noise conditions. A baseline resolution no worse than  40 eV is achieved on the thin detectors, which is used to study the low energy excess. A recoil threshold as low as 200 eV is expected. They are only protected by an inner 1" hermetic passive cooper shielding. The spectrum shows the single-scatter events observed by one 4 mm sapphire detector in the MINER stack of detectors. These large diameter (3") are not housed inside any active donut veto, unlike the germanium coin

with full hermetic shielding, the results of which have been presented in the past but not yet published due to the trigger threshold limitations in the earlier DAQ. The current DAQ operates in a triggerless mode and is thus not limited by any artificial trigger thresholds, although it comes at the expense of much more demanding resource requirements for data storage and the analysis pipeline.

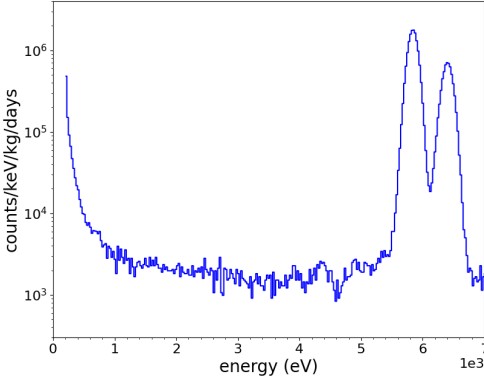
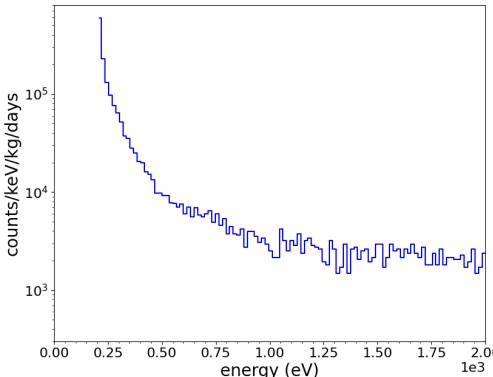

Figure 6: **(left)** MINER: The spectrum obtained from a 4mm thick sapphire detector with the [55] calibration lines, **(right)** The low energy excess. These events were obtained using a triggerless DAQ followed by software trigger for pulses.

### 2.1.4 NUCLEUS

*Section editor: Johannes Rothe (johannes.rothe@tum.de)*

This section describes an unshielded run of the first 1g-prototype target detector developed for the NUCLEUS experiment. The experimental run presented at the EXCESS workshop is described in [45] with the data and results on light DM published in [46]. It is presented as "Prototype Run 1" in [47].

**Detector concept and setup**     The NUCLEUS collaboration aims to detect CEvNS at a nuclear reactor using arrays of gram-scale cryogenic calorimeters [8]. The first prototype target detector shown in Fig. 7, using a cubical $Al_2O_3$ target of 5 mm side length (0.49 g mass), was operated at an unshielded facility at MPP (Max Planck Institute for Physics) Munich in February 2017. The detector uses a tungsten thin-film TES with aluminum phonon collectors, and a read-out chain based on a DC-SQUID amplifier. The sensor technology is very similar to and based on that of the CRESST experiment.
The detector holder consists of a copper plate, a bronze clamp and four sapphire spheres. The detector rests on three sapphire spheres and is clamped from the top via the fourth. The clamp also carries Cu-kapton-Cu traces which provide electrical and thermal connections via aluminum and gold wirebonds. The cube is otherwise unshielded and directly faces the cryostat vessels (the innermost being a copper shield at mixing chamber temperature).
 The dilution refrigerator hosting the experiment reached a base temperature of 11 mK. The TES was operated with a bias current of 1 $\mu$A and stabilized at its transition temperature of 22 mK with a small current through a resistive heater consisting of a small gold film deposited directly on the crystal.

**Data acquisition and processing**     Energy calibration of the detector was provided by a $^{55}$Fe source consisting of a metal stripe implanted with the isotope and covered by a kapton tape. The source delivered a rate of around 0.07 Hz of $^{55}$Mn K$_\alpha$ and K$_\beta$ x-rays at 5.9 keV and 6.5 keV.

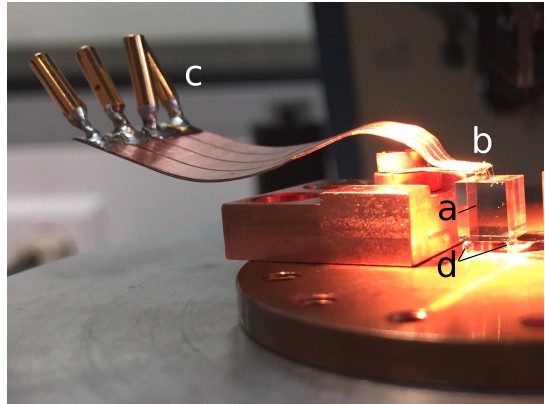
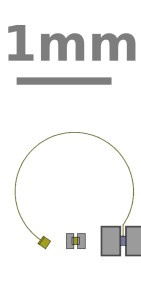

Figure 7: Left: NUCLEUS 1g-prototype setup: a) $Al_2O_3$ 5 mm cube used as a target; b) clamp holding the detector via a 1 mm $Al_2O_3$ sphere, with glued Cu-kapton-Cu bondpad for electrical and thermal connections; c) contacts for heater and bias lines; d) 3 $Al_2O_3$ spheres glued on a copper plate to support the target from below. Right: TES sensor layout used on the Nucleus 1g-prototype. Aluminum layers are shown in grey, tungsten in blue, gold layers in yellow. Left to right: thermal link bond pad, ohmic heater, TES sensor. Figures from Ref. [47].

Energy reconstruction was performed using two complementary methods: an optimum filter (for best energy resolution in the linear range) and a truncated fit method (to extend energy reconstruction into the saturation regime). The optimum filter was applied for events up to 600 eV, for which an undistorted pulse-shape following a model of two exponential components (as introduced in [48]) was observed. Higher energies are reconstructed at lower energy resolution by fitting a pulse template only to those samples of a pulse trace that fall within the linear response range of the detector (truncated fit, described e.g. in [32]). In this way, energy reconstruction could be performed beyond the linear range (up to 12 keV), which is necessary for calibration with the $^{55}$Fe source. The energy resolution found with the optimum filter method is $3.84 \pm 0.16$ eV. The trigger threshold of the detector was set to 19.7 eV.

The final energy spectrum of the 5.31 hour data acquisition was derived using several event selection criteria. In the first step, periods of unstable detector temperature were manually identified using saturated pulses and removed. This reduced the live time of the detector to 3.26 hours. Two energy-independent cuts were used against artifacts and mis-reconstructed events: a cut on pulse decay time removes heater pulses and mis-reconstructed saturated pulses, and a cut on the baseline slope removes SQUID resets and pile-up events. The cuts were set loosely so as to not affect the physical event population. In consequence, removed events are counted as dead time, further reducing the live time to 2.27 hours. This yields a final effective exposure time for the measurement of 0.046 g day.

**Energy spectrum from the 1g-prototype**    The final observed energy spectrum (Fig. 8) contains the calibration lines at 5.9 keV and 6.5 keV, a flat background (attributed to environmental gamma radiation) of around $6 \cdot 10^5$ count /keV /kg /day and a sharp rise in event rate below $\sim 200$ eV energy of currently unknown origin.

Subsequent to the first experimental run in February 2017, similar detectors were operated, proving out the holding and cryogenic veto concept of NUCLEUS [47]. "Prototype Run 2" featured a different TES design with a better energy resolution, a new silicon holder and calibration as well as background data-sets. The low-energy rise is present in the background dataset and therefore unrelated to the calibration source. "Prototype Run 4" operated for the

first time the "inner cryogenic veto" of NUCLEUS: flexible silicon wafers equipped with TES and holding the target cube. Operated in anticoincidence, these detectors reduced the low-energy event rate observed in the target. A sharp rise in the event rate remained below around 100 eV.

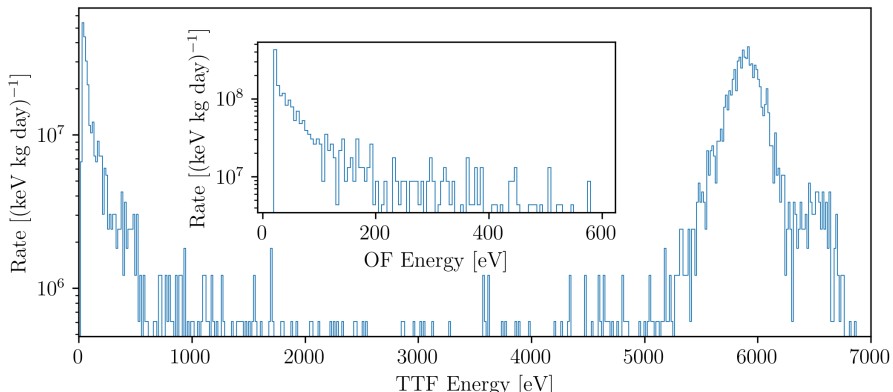

Figure 8: Final energy spectrum of the NUCLEUS-1g-prototype 2017. Main frame: complete energy range up to the $^{55}$Fe calibration lines, reconstructed with the truncated template fit (TTF). Inset: zoom on the low-energy region (19.7 eV - 600 eV), reconstructed with the optimum filter (OF). Figure from Ref. [47].

**Discussion**  The measured energy spectra were obtained in unshielded surface runs of detector prototypes for NUCLEUS. Backgrounds induced by known processes may explain the rising event rate at low energies. The NUCLEUS collaboration is working towards operating similar detectors in the complete experimental setup, including a passive shielding composed of lead and polyethylene, a high-efficiency muon veto and several cryogenic anticoincidence detectors. This setup will be commissioned at a shallow underground site at TUM. These measurements together with background simulations performed for the full setup will allow a comprehensive investigation of the origin of background events below 100 eV.

### 2.1.5 SuperCDMS - HVeV

*Section editors: Belina von Krosigk (belina.krosigk@kit.edu), Valentina Novati (valentina.novati@northwestern.edu)*

This section describes both the cryogenic bolometers and the data acquired in the two currently published SuperCDMS HVeV science runs: Run1 [11] and Run2 [12]. The respective detectors feature an eV-resolution and are sensitive to energy depositions as low as $\sim 1$ eV.

**Detector concept and setup**  Figure 9 shows the HVeV detectors used in Run1 and Run2. The detector absorbers are chips made of 0.93 g $(10 \times 10 \times 4)$ mm$^3$ silicon. Two channels of QETs are patterned on the top surface of the chips and act as athermal-phonon sensors. A different mask design is used in the two detectors, the second design showing an improvement of the energy resolution and an increase of the dynamic range for the Run2 detector [51]. An aluminum grid is deposited on the back of the detectors to apply a bias and enhance the signals thanks to the NTL effect. The silicon crystal substrates are held between two printed circuit boards (PCB) that provide the thermal and electrical contact for the devices. The PCBs

---

[3]Reprinted from [49], with the permission of AIP Publishing.

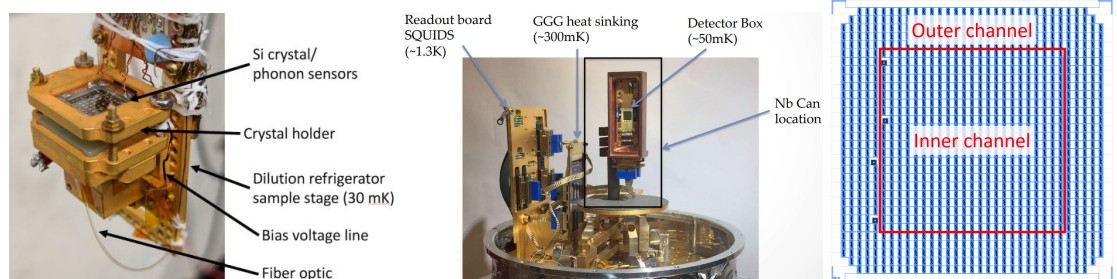

Figure 9: **Left:** SuperCDMS-HVeV Run1 detector mounted on the mixing chamber stage of a dilution refrigerator with a fiber optic to illuminate the detector from below. **Center:** SuperCDMS-HVeV Run2 detector installed in an adiabatic demagnetization refrigerator. **Right:** Drawing of the phonon sensor mask of the HVeV Run2 detector. Two channels with the same area are visible and their contacts highlighted with darker squares. Figures and captions from Refs. [49, 50][3].

were held together with four springs that, in the case of the Run2 detector, exercise a force of $50 - 70$ grams in each corner during operation. During Run1 the clamping was not measured but set to "finger tight". In each case, the detectors were enclosed in a light-tight copper holder.

The detectors can be operated both without (0V) and with (HV) a voltage bias applied on the electrodes: in the first case only the phonon signal generated by the event is detected, in the second case the semiconductor electron-hole pairs are drifted across the crystal amplifying the initial phonon signal. During both the Run1 and Run2 science exposure, the detectors were operated in HV mode: a bias of $-140$ V was applied on the Run1 detector after pre-biasing for five minutes to $-160$ V, and lower biases of 60 V and 100 V were used for the Run2 detector after pre-biasing for up to an hour to a voltage between 140V and 220V.

Two above-ground runs were performed with these detectors: (1) Run1 was performed at Stanford University (Stanford, CA) in a dilution refrigerator; (2) Run2 was performed at Northwestern University (Evanston, IL) in an Adiabatic Demagnetization Refrigerator (ADR). During Run1 the detector was operated at $33 - 36$ mK, and during Run2 it was stabilized at $50 - 52$ mK. No dedicated external shielding or veto systems were used in either of the two measurements. Only a secondary RF-sensitive detector was also present in Run2 but its performance was poor because its transition temperature was close to the ADR stabilized temperature.

**Data acquisition and processing** The data were acquired with a sampling frequency of 1.25 MHz (1.51 MHz) for Run1 (Run2). The Run1 data were triggered with a shaped pulse—sum of the two QET channels through a shaping amplifier. The Run2 data were taken continuously and triggered offline with a matched filter trigger. For both runs the amplitude of each event was calculated with an optimum filter.

The data were calibrated with a room-temperature laser with a wavelength of 650 nm for Run1 and 635 nm for Run2. The light signal was directed to the detector's center with an optical fiber. The fiber was pointed to the back side of the detector (shining onto the aluminum electrode) in Run1 and to the TES sensor in Run2. During the laser calibration, the detectors are illuminated by bursts of photons with an average photon number between 0.5 and 4. Single electron-hole pairs, corresponding to individual photons, are used to calibrate the detector and evaluate non-linearity at higher energies. A baseline resolution of 14 eV [49] and 2.7 eV [51] was achieved respectively during Run1 and Run2, where the energy resolution is expressed in total phonon energy and is independent of the applied voltage.

Live-time selections and data quality cuts were applied to these data. During Run1 data,

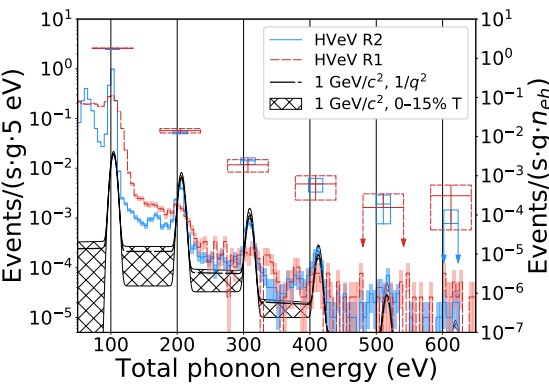

Figure 10: Energy spectra acquired during SuperCDMS-HVeV Run1 (R1) and Run2 (R2). An additional point is added above each electron-hole pair peak to highlight the event rate contained in a $3\sigma$ window around the peak (corresponding to the counts in the peak). Each point has a $3\sigma$ uncertainty on the number of counts. The black curve represents a DM-electron scattering model with DM form factor $F_{\mathrm{DM}} \propto 1/q^2$ and a DM mass of $1\,\mathrm{GeV}/c^2$ for an impact ionization of 2% and for a charge trapping of 11%. The uncertainty considers the trapping varying in the range 0 - 15%. Figure and caption from Ref. [50].

time periods with high noise and leakage where removed from the science data. Common to both runs, time periods were not considered in the analysis when the temperature was not stable and the detector was affected by high trigger rate due to noise or burst events. The event-quality cuts used for both runs ensure that the pulse shape of the events is similar to the laser pulse template, that the working point of the detector—which influences the detector gain—is stable and that the pulse position is correctly aligned with the trigger. Concerning Run2 data, a veto cut from a secondary detector mounted on the same holder was also applied.

During Run1 an exposure of 0.49 gram day was acquired, and 1.2 gram day were collected during Run2. The region of interest was set to $0.5-9$ electron-hole (e-h) pairs for Run1 and to $50-650$ eV total phonon energy for Run2.

**Energy spectrum from Run1 and Run2** The peaks visible in both Run1 and Run2 spectra correspond to the detection of single electron-hole pairs. The fill-in between the peaks may be caused by charge trapping and impact ionization [52]. An additional peak at around half of one-electron-hole pair is present in the Run2 data, which is due to charge trapping on the lateral surfaces of the silicon absorber in the current interpretation. As was shown in Eq. 1, the total phonon energy $E_{\mathrm{heat}}$ that is measured for a single particle interaction is the sum of the primary recoil energy $E_R$ of the interaction and the energy produced from the e-h pairs drifting in the electric field. In case of electron recoils all of the primary recoil energy is effectively converted into e-h pairs. While Eq. 2 provides a good description of the expected number of e-h pairs at high energies – where $\epsilon_{eh}$ is observed to be constant with a value of $\sim 3.7-3.8$ eV for silicon – it breaks down at energies as low as the ones measured with HVeV [53]. In this paper, the total phonon energy measured in the HVeV devices is converted into electron equivalent energy depositions (see Fig. 20) using Eq. 2 which is thus only to be considered a first order approximation.

The energy spectra observed in HVeV Run1 and Run2 are shown in Fig. 10. Periods of unstable environmental conditions (high voltage, temperature) were removed or corrected for and various event-based selection criteria were applied to the data to identify single pulses induced by particle interactions inside the target material. The energy and start time of all pulses

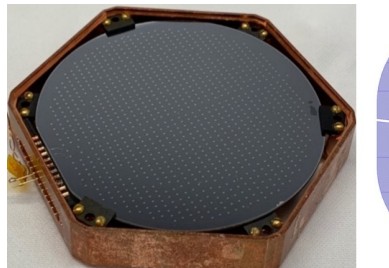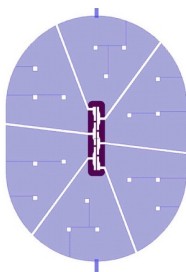

Figure 11: Left: A picture of the CPD installed in a copper housing. The instrumented side is shown facing up. Right: The design of the QETs used for the detector (blue: Al fins; purple: W TES). Figure and caption from Ref. [55][4].

were reconstructed using an optimal filter (OF) algorithm [51, 54] and a 650 nm (635 nm) laser was used to calibrate the Run1 (Run2) data. Both final spectra feature the quantized nature of the initial signal, where the first (second, third, ...) peak refers to one (two, three, ...) electron-hole pair created. The fill-in between the peaks is largely due to impact ionization and charge trapping [51].

A current hypothesis suggests that a large fraction of the events observed in the spectra shown in Fig. 10 is due to luminescence induced in material in the direct vicinity of the target material. The SuperCDMS collaboration is testing this hypothesis.

### 2.1.6 SuperCDMS - CPD

*Section editor: Samuel Watkins (samwatkins@berkeley.edu)*

This section describes the Cryogenic PhotoDetector (CPD) used and the data acquired during the SuperCDMS-CPD DM search [13, 55].

**Detector concept and setup**  The substrate of the detector is a 10.6 g Si wafer of 1 mm thickness and 45.6 cm$^2$ surface area. On one side of the wafer, a single uniformly-distributed channel of QETs was deposited, which operate at a superconducting critical temperature of $T_c = 41.5$ mK. The other side of the wafer is not instrumented and unpolished. The wafer itself is held in a copper housing by six cirlex clamps.

The DM search was carried out at the SLAC National Accelerator Laboratory for an exposure of 9.9 g days in a cryogen-free dilution refrigerator with a base temperature of 8 mK. The SLAC facility is located at the surface and had minimal shielding. To calibrate the detector, a collimated $^{55}$Fe x-ray source, along with a 38 $\mu$m layer of Al foil, was placed incident on the noninstrumented face of the detector to provide peaks at 1.5, 5.9, and 6.5 keV. The detector installed in its copper housing and the QET design are shown in Fig. 11.

**Data acquisition and processing**  The data for the DM search were acquired using a FPGA triggering algorithm based on the optimal filter (OF) formalism, which acts on a downsampled version of the raw data stream (downsampled from a digitization rate of 625 kHz to 39 kHz). The trigger threshold was set at 4.2$\sigma$ above the baseline noise level, and events with OF amplitudes above this level were saved at the full digitization rate. An offline OF is then applied to the saved data to extract OF amplitudes from each event to be used as the reconstructed energy estimator. The baseline energy resolution of the offline OF is $\sigma_E = 3.86 \pm 0.04$ (stat.)$^{+0.19}_{-0.00}$ (syst.) eV. The energy calibration of the offline OF only applies

---

[4]Reprinted with permission from Ref. [55], with the permission of AIP Publishing.

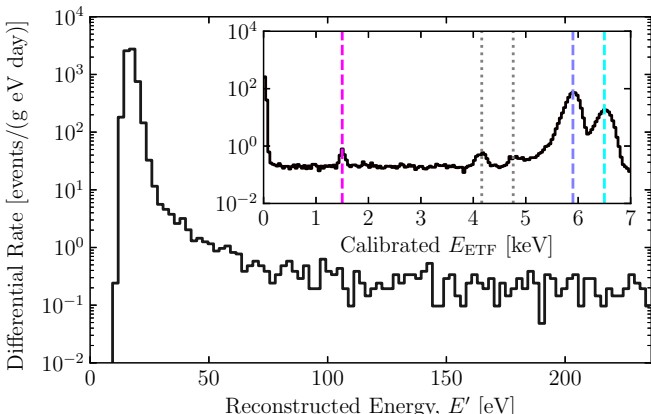

Figure 12: Measured energy spectrum in the DM-search ROI for the full exposure after application of the quality cuts. The data have been normalized to events per gram per day per eV and have been corrected for the event-selection efficiency, but not the trigger efficiency. The inset shows the calibrated $E_{\mathrm{ETF}}$ spectrum up to 7 keV, noting the locations of the different spectral peaks. The known values of the dashed lines are 1.5, 5.9, and 6.5 keV for the Al fluorescence (pink), $^{55}$Fe K$_\alpha$ (blue), and $^{55}$Fe K$_\beta$ (cyan) lines, respectively. The two dotted gray lines between 4 and 5 keV in calibrated $E_{\mathrm{ETF}}$ are the Si escape peaks [56]. Figure and caption from Ref. [13].

to the DM region of interest (ROI) below 240 eV, as there were nonlinear effects due to pulse saturation at higher energies. For the EXCESS Workshop, a calibrated spectrum using a pulse integral energy estimator was also supplied, which provides energies up to 7 keV. However, the baseline resolution of this energy estimator is about a factor of 10 worse than the OF energy estimator used in the ROI.

Energy-independent data quality cuts were applied to the ROI energy spectrum, consisting of a prepulse baseline cut and a goodness-of-fit cut that together had a 88.7% total signal efficiency.

**Energy spectrum from the DM search**    The observed energy spectrum in the ROI is shown in the main plot of Fig. 12. Above 100 eV the spectrum consists of a flat background of $2 \cdot 10^5$ count/keV/kg/day, which is attributed to Compton scattering of the gamma ray background. Below 100 eV, the spectrum rises exponentially above the flat background. Below 30 eV, the spectrum rises at a steeper exponential, which could be due to random noise fluctuations above the trigger threshold.

**Discussion**    The origin of the excess observed between 30 and 100 eV is unknown. Possible sources include Cherenkov interactions, transition radiation, other low energy interactions with high energy particles, neutrons, EMI signals, or stress microfractures from the clamping of the detector. It has been shown that Cherenkov interactions and transition radiation could only account for up to 10% of the observed background [57], thus these cannot fully explain the observed excess. SuperCDMS is analyzing data obtained from operating this detector in an underground setting to study the other background candidates. There are also plans to test clamping schemes designed to reduce stress microfractures, with a concurrent goal of reducing sensitivity to pulse-tube cryocooler vibrations.

In this section, we described the observations of low energy excess signals in cryogenic detectors. We continue in the following section with observations from CCD detectors.

## 2.2 CCD detectors

Charge Coupled Devices (CCDs) are used in many scientific applications. A CCD consists of a semiconductor substrate (usually silicon, though germanium CCDs are under development [58]) with a thickness of up to 1 mm, patterned with an array of pixels and depleted of free charges using an applied bias voltage. Electron-hole pairs generated in the substrate are collected in the pixels and shifted to readout transistors, which give a measurement of the charge deposited in each pixel. In the context of DM detection, CCDs are able to measure DM interactions that deposit energies as small as the semiconductor band-gap, i.e., of the order of a few eV, thus enabling the detection of MeV-scale DM [59, 60]. In contrast with cryogenic detectors, CCDs for DM detection are operated at relatively high temperatures, between 100 to 150 K. The output signal of a CCD is proportional to the charge collected in each pixel, with the charge resolution being limited by electronic noise in the readout transistor. The energy resolution is additionally subject to the process of converting energy to electron-hole pairs; on average, each 3.8 eV of electron recoil energy produces an additional electron-hole pair, but the precise number is subject to fluctuations that can be quantified with a Fano factor [53]. However, for the shown energy spectra, the conversion factor of 3.8 $eV_{ee}$ per electron-hole pair is used. This conversion is model-dependent, i.e. assumes that the type of recoil was an interaction with the electrons of the target material, not with the atomic nuclei.

**Skipper-CCDs** are CCDs with a special readout transistor that allows for multiple nondestructive measurements of the same charge packet [61]. By measuring each pixel $N$ times, the readout noise can be reduced by a factor of $\sqrt{N}$, to the point where single elementary charges can be clearly resolved.

All CCDs currently used in DM experiments are made of high-resistivity silicon and were designed by the LBNL Microsystems Laboratory (MSL) [62] and fabricated at Teledyne-DALSA. In the following sections, we describe the CCD and Skipper-CCD measurements that were presented at the EXCESS workshop.

### 2.2.1 DAMIC

*Section editors: Daniel Baxter (dbaxter9@fnal.gov), Alvaro Chavarria (chavarri@uw.edu)*

In this sections we explain the results of the **DA**rk **M**atter **I**n **C**CDs (DAMIC) experiment at SNOLAB, which is the first DM detector to utilize a multi-CCD array [63].

The detector is located 6800 ft underground (6000 m.w.e.) in SNOLAB underground laboratory [64] and surrounded by 20 cm of lead plus 42 cm of high-density polyethylene passive shield on all sides to eliminate external background gammas and neutrons respectively. Remaining background events come from the intrinsic radioimpurity of the detector materials themselves. The CCDs and copper IR shield are held at 140 K using a commercial Cryomech cryocooler unit. Each CCD is instrumented using a single Kapton flex cable, which exits the passive shielding through a vertical channel in the lead where it feeds through a vacuum interface board (VIB) to the CCD controller, a Monsoon system developed for the Dark Energy Camera [65, 66].

The CCDs installed in DAMIC's most recent run pre-date the application of Skipper amplifier technology for DM searches, but are still able to achieve 1.6 $e^-$ (6 $eV_{ee}$) resolution in a single pixel measurement using the correlated double sampling technique [69]. These CCDs are calibrated in-situ using a red (780 nm) light-emitting diode inside the vacuum cryostat [63]. The detector leakage current was measured to be 2–6 $\times 10^{-22}$ A/cm$^2$ (or 600–1680 $e^-$/g-day) [70].

The data used in the most recent results were taken with seven 4k×4k pixels (6g) CCDs between September 2017 – December 2018, consisting of a total exposure of 11 kg-days.

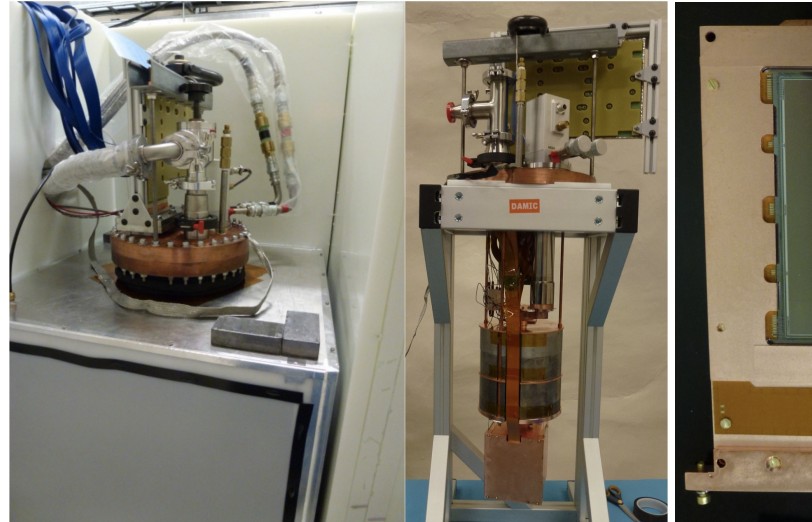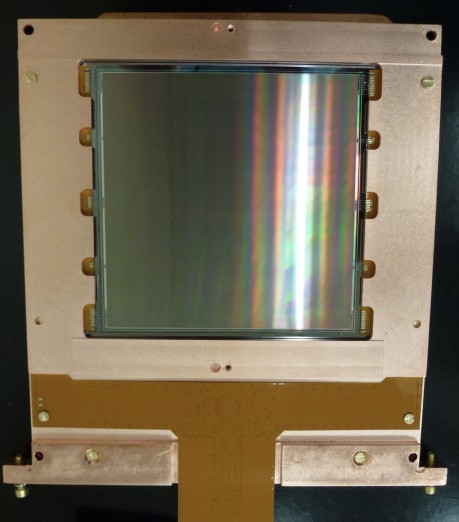

Figure 13: **From left to right:** Photographs of the DAMIC detector at SNOLAB show-
ing the sealed copper cryostat inside its radiation shield, with electronics and service
lines connected to the feedthrough flange, of the cryostat insert showing the Kapton
flex cables running from the CCD box to the vacuum interface board along the chan-
nel in the internal lead shield, and of the DAMIC CCD module housing a 4k x 4k pixel
CCD. Figures from Refs. [67, 68].

One of these CCDs is housed in a copper module that was electroformed at Pacific Northwest
National Labs [71] and sandwiched between ancient lead bricks, resulting in a background
rate of 3.1 count/$keV_{ee}$/kg/day (between 2.5–7.5 $keV_{ee}$), the lowest of any silicon detector
to date [67]. The requirement that the expected number of events from noise is <0.1 in the
exposure sets the analysis threshold of 50 $eV_{ee}$. Higher energy events above 6 $keV_{ee}$ are used to
construct a background model between 0.05–6 $keV_{ee}$ in both energy and pixel spread, which
is positively correlated with event depth. This model is found to be in excellent agreement
with data above 200 $eV_{ee}$ [67].

Between 50–200 $eV_{ee}$, a statistically significant ($p$-value of $2.2 \times 10^{-4}$) excess of $17.1 \pm 7.6$
events is observed above the background model expectation [73]. These events are consistent
with a bulk spectrum decaying exponentially with a decay constant of $(67 \pm 37)$ eV [67]. To
verify that this excess is indeed robust, Skipper CCDs have been installed in the DAMIC at
SNOLAB detector, in collaboration with the SENSEI and DAMIC-M, allowing a measurement
of the same, well-characterized background environment with lower threshold.

### 2.2.2 SENSEI

*Section editors: Rouven Essig (rouven.essig@stonybrook.edu), Sho Uemura
(meeg@slac.stanford.edu)*

The SENSEI collaboration performed a DM search at a shallow underground site, with
sensitivity to events creating 1–4 electron-hole pairs [74].

The experiment was operated in the MINOS cavern at Fermilab, ~104 m (225 mwe) [75]
underground. One Skipper-CCD was packaged and installed as shown in Figure 15. The
Skipper-CCD was shielded with lead both inside and outside the vacuum vessel, in a
non-hermetic configuration that resulted in a background radiation rate of
~3370 count/keV/kg/day in the range from 500 eV to 10 keV energy. The Skipper-CCD
was maintained at a temperature of 135 K using a commercial Cryomech cryocooler unit.

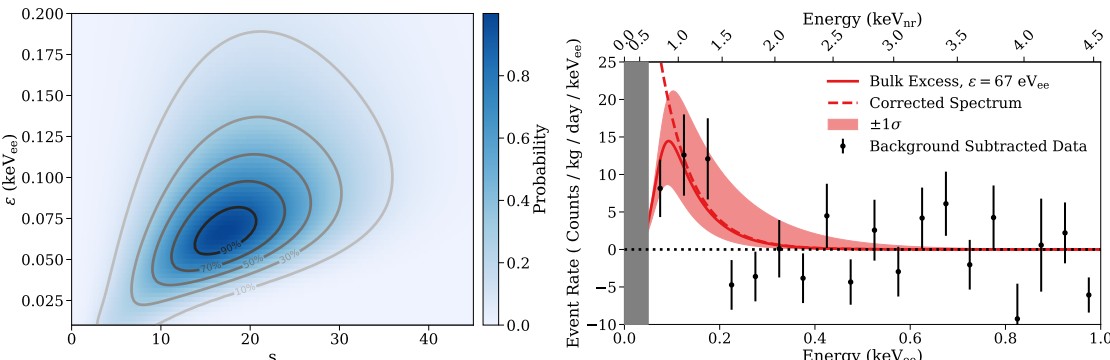

Figure 14: **Left:** Fit uncertainty in the number of excess signal events over the background model (s) and characteristic decay energy ($\epsilon$) of the generic exponential signal spectrum. The color axis represents the $p$-value from likelihood-ratio tests to fit results with constrained s and $\epsilon$. **Right:** Energy spectrum of the best-fit generic signal (red lines) overlaid on the background-subtracted data (markers). The subtracted background model is non-trivial and explained in detail in Ref. [67]. Both the fit spectrum that includes the detector response (solid line) and the spectrum corrected for the detection efficiency (dashed line) are provided. The ionization efficiency used to construct the equivalent nuclear recoil energy (keV$_{nr}$) shown on the top x-axis is taken from the direct calibration performed in Ref. [72]. Figures from Ref. [67].

The Skipper-CCD has 886 columns and 6144 rows of pixels, each of dimensions 15 $\mu$m by 15 $\mu$m, and a thickness of 675 $\mu$m, for a total active mass of 1.926 grams. The active area was divided in quadrants, and each quadrant was read out through a Skipper amplifier at a corner of the Skipper-CCD. The charge measurement was calibrated for each quadrant using Gaussian fits to the discrete charge peaks. Two quadrants performed well in all respects, with readout noise of 0.146 e$^-$ and 0.139 e$^-$. One quadrant was inoperable, and its data was discarded. Another quadrant (with a readout noise of 0.142 e$^-$) had an excess of 1 e$^-$ events attributed to a light leak; its data was discarded for the 1 e$^-$ and 2 e$^-$ analyses, but included (after removing the portion of the quadrant with the largest excess) in the 3 e$^-$ and 4 e$^-$ analyses.

The data-collection cycle consisted of a 20-hour exposure followed by a full readout of the CCD (300 measurements per pixel, requiring 5.153 hours to read out the active area). The data from each such cycle comprises an "image," and 22 such images were included in the blinded dataset, in addition to 7 commissioning images that were used to develop the analysis. The total blinded exposure (before cuts) was 19.93 g-day for the 1 e$^-$ and 2 e$^-$ analyses, and 27.82 g-day for the 3 e$^-$ and 4 e$^-$ analyses.

Analysis cuts were applied to reject regions of each image that contained large amounts of charge from high-energy events and regions where excess charge could be expected (at the locations of known CCD defects, near high-energy events, in regions with an excess of other events). The 1 e$^-$ and 2 e$^-$ analyses searched for single-pixel events; the 3 e$^-$ and 4 e$^-$ analyses searched for contiguous clusters of nonempty pixels. The effective exposure for each analysis was corrected for the efficiency of the cuts and the probability that a DM event generating the given amount of charge would diffuse into a configuration accepted by the analysis (a single pixel for 2 e$^-$, a contiguous cluster of pixels for 3 e$^-$ and 4 e$^-$).

A (1 e$^-$, 2 e$^-$, 3 e$^-$, 4 e$^-$) pixel was defined to have a *measured* charge in the range ((0.63,1.63], (1.63,2.5], (2.5,3.5], (3.5,4.5]) e$^-$, respectively. The number of 1 e$^-$

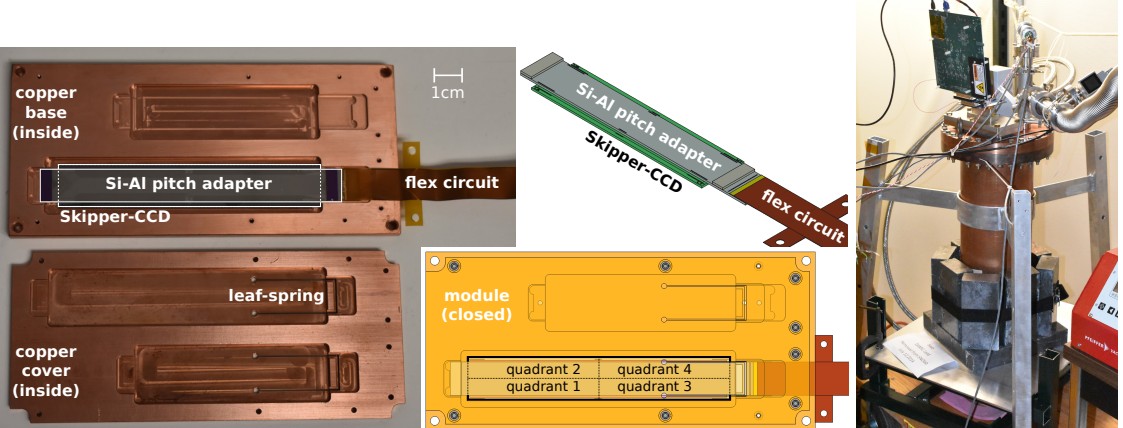

Figure 15: The CCD used in the 2020 SENSEI measurement was glued to a silicon-aluminum pitch adapter, which in turn was laminated with a copper-Kapton flex cable; this CCD module was then placed in a copper tray (left). Figure from Ref. [74][5]. The tray was installed in a vacuum vessel with a lead shield (right), underground in the MINOS cavern at Fermilab.

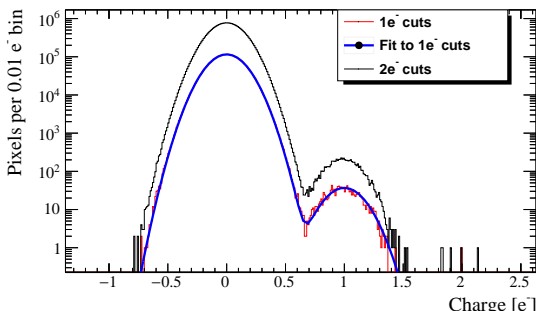

Figure 16: Spectrum of measured pixel charge in SENSEI, after the event selection cuts for the $1\,e^-$ and $2\,e^-$ analyses. A double-Gaussian fit is shown for the $1\,e^-$ selection. There were no $3\,e^-$ or $4\,e^-$ events. Figure from Ref. [74][6].

events in the data was corrected for misclassification using a Gaussian fit as shown in Fig. 16 (i.e., a few empty pixels would be measured to have a charge consistent with a $1\,e^-$ event).

These data led to a measurement of the lowest rates in silicon detectors of events containing $1\,e^-$, $2\,e^-$, $3\,e^-$, or $4\,e^-$. The measured rates were then converted to constraints on DM that produce such events [74]. The computed limits on the $1\,e^-$ event rate accounted for the contribution of "spurious charge" [76], which was measured separately.

Ref [76] contains a detailed study that disentangles different contributions to the $1\,e^-$ events observed with the Skipper-CCD.

**Discussion** The $1\,e^-$-event rate (after all analysis cuts and after subtracting the spurious charge contribution) corresponds to a rate of $(450 \pm 45)$ events/g-day. Intriguingly, removing the lead shielding surrounding the vacuum vessel, which produced a larger measured value for the high-energy event rate, led to an increase in the measured $1\,e^-$-event rate [74]. This suggests an environmental origin for the $1\,e^-$ events. Likely mechanisms for generating these $1\,e^-$ events include Cherenkov radiation by high-energy events interacting in the silicon of the Skipper-CCD and radiative recombination of electron-hole pairs that are produced in a

thin highly-*n*-doped region on the backside of the Skipper-CCD [57]. A detailed simulation to check this hypothesis is in progress.

### 2.2.3 Skipper CCD running above ground at Fermilab

*Section editor: Guillermo Fernandez Moroni (gfmoroni@fnal.gov)*

In this section we discuss recent results from a Skipper-CCD operated above ground.

Figure 17 (a) shows the experimental setup used to test the performance of the Skipper above ground at Fermilab in 2021. Some of the main components are labeled. One Skipper-CCD (shown in Fig. 17 (b)) was operated at a temperature of 140 K using a Sunpower cryocooler [77]. The CCD is glued to a silicon substrate that sits on a copper tray for mechanical support as well as thermal connectivity. The CCD is placed in an extension of the dewar that fits inside a lead cylinder. A lead cap on top of the sensor (inside the dewar extension) completes the lead shield of two inches of thickness around the device. There was no radiopurity selection of materials inside the shield. The CCD has 6144 columns by 1024 rows with pixels of 15 $\mu$m by 15 $\mu$m with a thickness of 675 $\mu$m. It is read by four amplifiers, one on each corner, using a Low Threshold Acquisition (LTA) controller [78]. Two quadrants presented larger readout noise and were not used for the analysis in the following sections. The sensor was operated at sub-electron noise by averaging 300 measurements of the charge in each pixel [79] and with a horizontal binning [69] of 10 columns. 3.21 days of data were collected from the active region of the sensor in continuous readout mode. Each output image of the sensor was taken every approximately 54 minutes. More about continuous readout mode can be seen in [74]. Columns of the CCD that presented high single electron rate (hot columns [69]) were eliminated from the analysis at an early stage. The remaining active mass of the sensor in use is 0.675 grams.

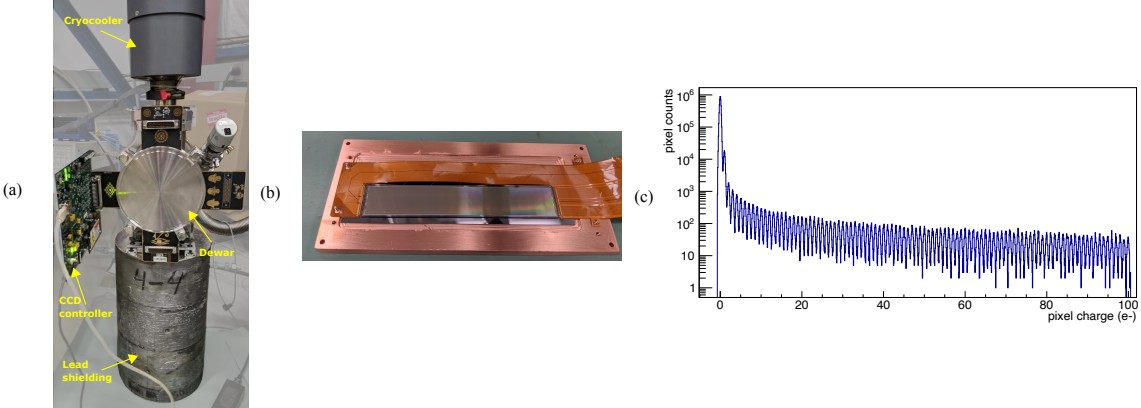

Figure 17: (a) Setup used for the Skipper-CCD experiment with a short description of the main components. (b) A picture of the CCD installed on the copper tray. An extra copper plate that covers the top part of the sensor is not presented in the image. (c) Histogram of pixels with charge up to 100 e$^-$ after calibration. Single electron discrimination is observed. Figures adapted from Ref. [80].

An absolute energy calibration of the sensor is performed using the electron counting capability. A histogram of the pixel values from the active region is produced as shown in Fig. 17(c). Each peak correspond to a discretized number of electrons in the pixel after the calibration and is fitted using a Gaussian distribution whose mean value is used to build a look-up

---

[6]Reprinted with permission from Ref. [74]. Copyright 2020 by the American Physical Society.

table (digital unit vs. electrons) to calibrate the sensor up to around $700\,e^-$. To calibrate the sensor at $2146\,e^-$, single pixel X-ray events with energy of 8.048 keV produced by the fluorescence of the surrounding copper material are also used. We assume an average energy deposition per collected electron of 3.75 eVee [81].

The readout noise of the sensor, evaluated as the standard deviation of the values of the empty pixels ($0\,e^-$ peak in Fig. 17(c)), is $0.165\,e^-$ and $0.167\,e^-$ for the two quadrants in use. The average single electron rate per pixel measured are $0.01\,e^-/\text{pix}$ and $0.009\,e^-/\text{pix}$ after binning in each quadrant. The energy resolution is less than $1\,e^-$.

Figure 18 is the measured spectrum of events after selection cuts without scaling by efficiency. Each energy bin is 100 eV wide and the first bin starts at 15 eV. The efficiency is almost constant in the energy range of the figure ( 60%). More details can be found in [80]. Although the first two bins of the spectrum show a slightly higher count rate, there is no evidence of a rapid increment of background events towards low energies. More studies are being carried out to get more details of the background behavior in this region beyond what can be stated as the current statistical limitation.

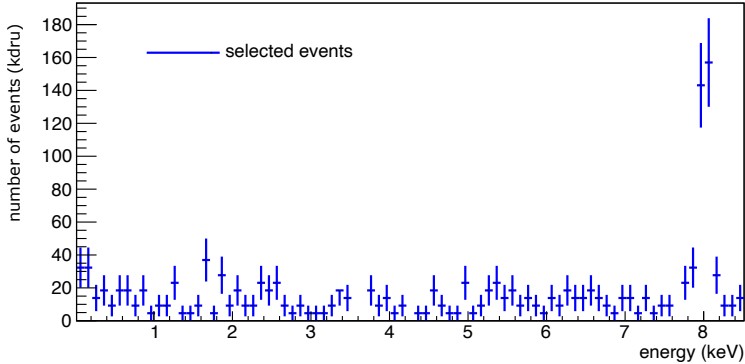

Figure 18: Measured spectrum of the Skipper-CCD experiment, after applying selection criteria.

In this section, we described the observations of low energy excess signals in CCD detectors. We continue in the following section with observations from detectors with gaseous targets.

## 2.3 Gaseous ionization detectors

Ionization detectors with a gaseous target are used by one of the contributing collaborations to carry out rare event searches.

**Spherical Proportional Counters (SPCs)** [7, 82–84] are gaseous detectors that record the ionization signal generated by incoming radiation. Incident particles interacting in the gas generate ion-electron pairs proportionally to the deposited energy. The released primary electrons will drift towards the central anode. As they do, they will increasingly diffuse with respect to each other the longer they drift, allowing for identification of surface events based on the spread of the primary electrons. Once they reach the intense electric field close to the anode, an avalanche process will release thousands of secondary ion-electron pairs per primary electron, allowing observation of events down to a single primary electron. The secondary ions will induce a current on the anode as they drift away from it, which is then integrated by the readout electronics and digitized.

In the following, we describe the NEWS-G experiment and its observation of a low energy excess.

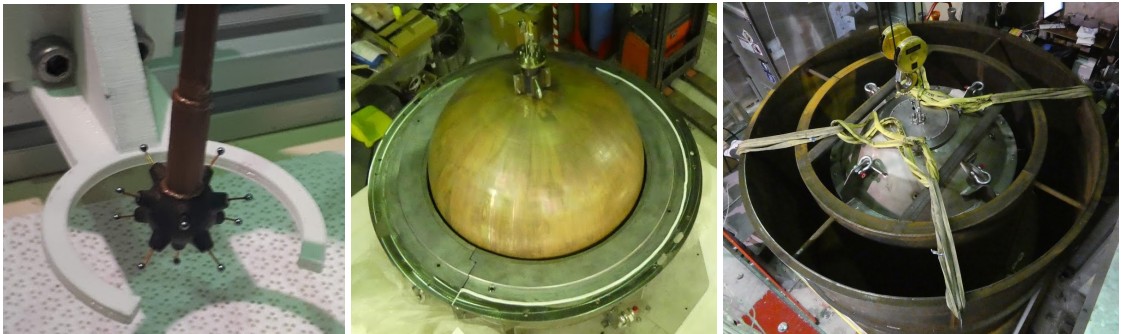

Figure 19: NEWS-G's S140 setup at LSM (2019). **From left to right:** ACHINOS anode; S140 sitting on the lower half of the lead shield; S140 in the closed lead shield, inside the open water shield.

### 2.3.1 NEWS-G

*Section editors: Francisco Vazquez de Sola (vazquez@subatech.in2p3.fr)*

**Detector concept and setup** The NEWS-G S140 detector, currently being tested at SNO-LAB [64], is a high purity copper (C10100) 140 cm diameter detector, with 500 μm of pure copper electroplated on the inside surface of the detector shell to attenuate the backgrounds both from the $^{210}$Pb contamination on the internal surface of the detector and $^{210}$Bi in the copper bulk [85]. The data described in this work was obtained during the commissioning at the Laboratoire Souterrain de Modane (LSM) [86] in 2019, under 4800 m of water-equivalent overburden. The detector was enclosed in 25 cm of lead, of which the internal 3 cm is archaeological lead, and an outer 34 cm thick water shield on the sides and 34 cm thick layers of HDPE above and beneath it. Images of the setup are shown in Fig. 19.

A new kind of anode, the ACHINOS [87, 88], was developed to accommodate the larger detector size. The one used, shown in Fig. 19, consists of a DLC-coated [89], 3D-printed, 1.6 cm wide nylon support, holding eleven 1.7 mm-diameter silicon anodes via 0.5 mm thick insulated wires, and connected to the grounded support rod through a 3D-printed 0.8 cm wide nylon tube covered in a layer (< 0.5 mm) of low-radioactivity araldite mixed with copper (30-35% w/w). The anodes were split into two readouts: the "North" channel, comprising the five anodes closest to the rod, and the "South" channel, comprising the six furthest. Only events reaching the South anode were kept, to avoid the field anisotropies close to the rod.

**Data acquisition and processing** From the commissioning runs, 156 hours of data were taken in 135 mbar of $CH_4$ (0.114 kg), 2030 V applied on the ACHINOS, fixed trigger conditions, and a sampling frequency of 1.04 MHz. The results discussed in this work represent South hemisphere data for only 21 hours from the whole dataset, for a total exposure of 0.0156 kg · days. A $\lambda = 213$ nm pulsed laser [90] was shined into the detector through a fiber feedthrough to monitor the stability of the gain and drift during physics runs by extracting photoelectrons from the S140 internal surface. Additional daily one hour low-intensity laser calibrations were performed to study the single electron response of the detector. At the end of the physics runs, eleven hours of data with $^{37}$Ar were taken under the same operating conditions to calibrate the fiducial volume associated with each ACHINOS channel and the attachment rate of primary electrons. Together with the low intensity laser data, this doubled as a measure of the mean ionization energy at 2.8 keV and 270 eV.

Events in the detector are identified by running a trapezoidal filter with the acquisition

software, triggering whenever a given threshold is reached, then storing a 8 ms window centered on the trigger time. The offline processing consists of a running average over 7 samples to remove high-frequency noise, a deconvolution of the single electron response function, and a cumulative integration. The resulting signal's amplitude is proportional to the energy deposited in the detector, and the risetime is correlated with the radial position of the initial interaction.

The larger diffusion of primary electrons in the S140, compared to previous smaller SPCs, increased the impact of low frequency noise on parameter estimation, particularly under $1 \, keV_{ee}$. Conversely, it allowed for a new analysis approach at very low energies. For events with low number of primary electrons, the ROOT TSpectrum peak searching algorithm [91,92] is used on the deconvolved signal, with each peak identified as the arrival of a primary electron to the anode. Then the number of peaks found is used to estimate the energy of the interaction, and the time separation between the first and last peak to determine statistically the radial position of the interactions.

The combination of the online trigger, TSpectrum peak-finding, and a threshold at half the mean avalanche gain resulted in a 50% detection efficiency for single electrons, approaching 100% quickly with more primary electrons. For the offline analysis based on the deconvolved signal, the baseline RMS was under 10% of the mean amplitude of a single electron signal; the threshold for peak identification was set at 6 times this value to optimize sensitivity while keeping a tolerable rate of false positives.

**Discussion**   The data being discussed is currently still being analyzed to produce a WIMP exclusion limit. As such, it is not ready to be published within this work. However, some preliminary results can still be discussed, notably on background rejection for single-electron data, where most 0.1 GeV WIMP recoils should fall.

The first source of low-energy background are so-called spurious pulses, believed to originate from detector electronics, and characterized by a pulse-shape that does not match those from laser-induced photoelectrons. These are rejected by a combination of two cuts. The first is based on the risetime of their raw pulse, shorter for pulses originating in the electronics than for pulses induced by the drift of secondary ions. The second is based on the relative signal between the south and north anodes. For this achinos configuration, secondary ions drifting away from the South anodes induce simultaneously a positive signal on that channel and a negative signal on the North channel of 20% of the amplitude of the positive signal. South spurious events do not induce any signal on the North channel, and so can be rejected. Both cuts together decrease the sensitivity to primary electrons by a relative 23%, but reject spurious pulses representing 62% of all single-peak data in the run.

The second source of observed low-energy background is correlated in time with high energy alpha events, primarily from $^{210}$Po 5.3 MeV decays in the copper shell. After each alpha, the rate of single electron events jumps up to 50 Hz, progressively going back to the baseline rate after a few seconds. The physical process behind this long electron tail, much longer than the electron drift time of only 1.3 ms, is not understood at this time, but they can be rejected by adding 5 s of dead time after each alpha event. This leads to an exposure loss of 13%, while rejecting alpha-correlated events representing 65% of all single-electron events.

Even after applying both selection criteria, a rate of 0.5 Hz of single electron events of unknown origin is still observed, orders of magnitude higher than the double-electron event rate in the data. Accounting for cut efficiencies, effective run time and fiducial volume, this is approximately 1 Hz of single-electron events in the whole sphere. Approximating single electron events as coming from a range of energies of $28 \, eV_{ee}$ (the mean ionization energy in the gas [93–95]), this is equivalent to $3 \cdot 10^7$ count/$keV_{ee}$/kg/day of target mass, or $6 \cdot 10^5$ count/$keV_{ee}$/m$^2$/day of detector surface. In the absence of an explanation for the origin

of this large rate, physics searches have been limited to signals with two electrons or more.

## 3 Comparison of the measured spectra

Table 1: Key properties of the measurements presented at the EXCESS workshop. First part contains the experiments shown in the Fig. 20a, second part corresponds to the Fig. 20b. The spectrum of the experiment in the third part is not shown in this work.

| Measurement | Target | Sensor | Exposure (kg days) | Operation Temperature | Depth (m.w.e.) |
|---|---|---|---|---|---|
| CRESST III DetA | 23.6 g CaWO$_4$ | Tungsten TES | 5.594 | 15 mK | 3600 (LNGS) |
| EDELWEISS RED20 | 33.4 g Ge | NTD | 0.033 | 17 mK | above ground |
| MINER Sapphire | 100 g Al$_2$O$_3$ | QET | 2.72 | 7 mK | above ground |
| NUCLEUS 1g prototype | 0.49 g Al$_2$O$_3$ | Tungsten TES | 0.0001 | 15-20 mK | above ground |
| SuperCDMS CPD | 10.6 Si | QET | 0.0099 | 41.5 mK | above ground |
| DAMIC | 40 g Si | CCDs | 10.927 | 140 K | 6000 (SNOLAB) |
| EDELWEISS RED30 | 33.4 g Ge | NTD, NTL amplification | 0.081 | 20.7 mK | 4800 (LSM) |
| SENSEI | 1.926 g Si | Skipper CCD | 0.0955 | 135 K | 225 (Fermilab) |
| Skipper CCD | 0.675 g Si | Skipper CCD | 0.0022 | 140 K | above ground |
| SuperCDMS HVeV Run 1 | 0.93 g Si | QET, NTL amplification | 0.00049 | 33-36 mK | above ground |
| SuperCDMS HVeV Run 2 | 0.93 g Si | QET, NTL amplification | 0.0012 | 50-52 mK | above ground |
| NEWS-G | 114 g CH$_4$ | SPC | 0.0156 | Room temperature | 4800 (LSM) |

After describing the individual observations of a low energy excess in Section 2, we proceed with a comparison of the presented data.

Table 1 contains an overview of some key properties of the measurements: The target mass and material, the sensor, exposure, operation temperature, and overburden. The measurements were taken in four underground laboratories and numerous above ground laboratories. The operation temperature is significantly different for different sensor concepts, ranging from several tens of millikelvin for sensors of cryogenic detectors, to $\mathcal{O}(100)$ K for CCD-based sensors and room temperature for gaseous ionization detectors. In total 5 different target materials were used in the measurements, with exposures up to 10 kg days, all observing a rising event rate at low energies.

In Fig. 20a and 20b we show selected recoil energy spectra that were discussed during the workshop. We separated the measurements according to their energy units: The CRESST, EDELWEISS RED20, MINER, NUCLEUS and SuperCDMS-CPD measurements are in units of total energy deposition, while the DAMIC, EDELWEISS RED30, SENSEI, Skipper-CCD and SuperCDMS-HVeV measurements are in units of electron equivalent energy, i.e. assuming that all incoming particles scattered off electrons in the detector material. It is important to note

that the conversion from electron equivalent to nuclear recoil units is possible and conversion factors are well studied for most detector materials used. However, without knowledge about the origin of measured signals, a comparison to the results of experiments which measure electron recoils and nuclear recoils on the same energy scale will hinge on the validity of the underlying interaction assumption. A more independent framework for comparison of all experiments is the matter of ongoing discussions within the workshop community. All spectra are scaled to count/keV/kg/day. This scaling is standard for rare event searches, as usually the sought-for signal scales with exposure. However, the scaling might be suboptimal to identify the origin of the excess, as the excess might very well not scale with exposure, but instead for example with surface or measurement time. For different variations of binning and display ranges, as well as other combinations of spectra, we refer the reader to the interactive visualization tools, hosted in the EXCESS workshop data repository [15]. Within the data repository the original data is available, and we encourage its usage for the creation of plots with alternative scaling that the community may wish to explore.

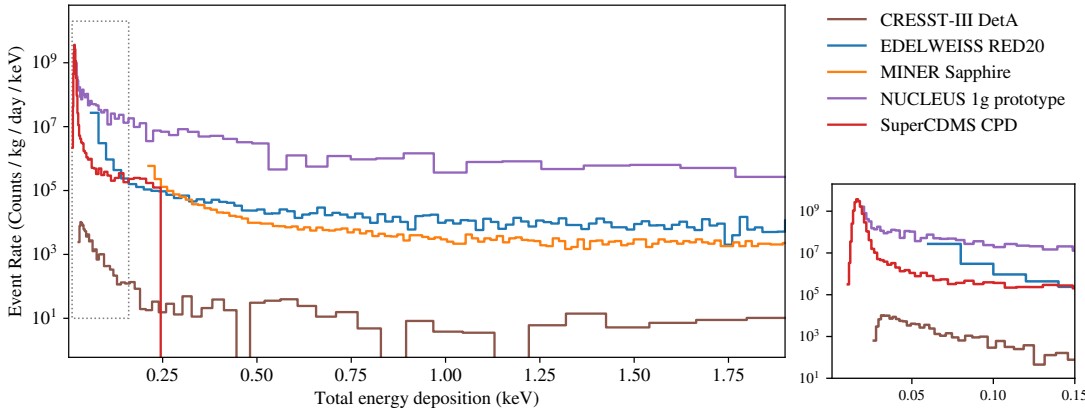

(a) Energy spectra of measurements with units of total energy deposition. The apparent peaks in the CRESST (SuperCDMS CPD) data at 30 eV (20 eV) are caused by the trigger threshold and discussed in the main text.

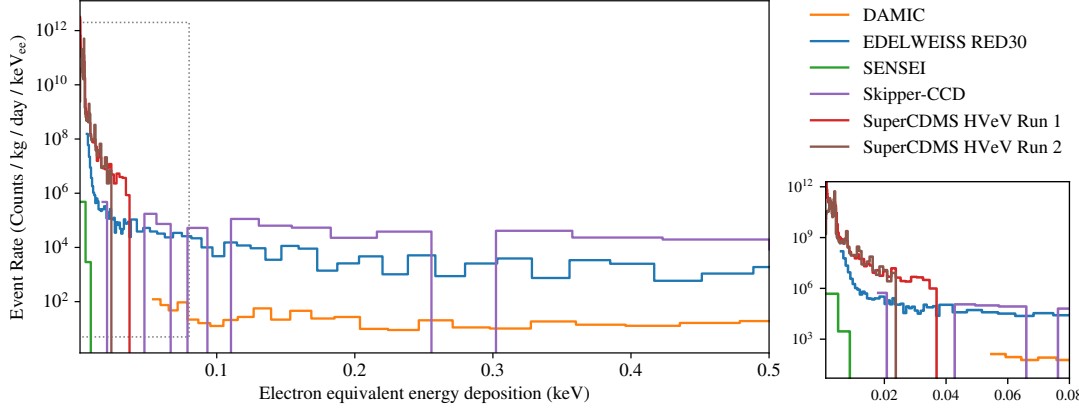

(b) Energy spectra of measurements with units of electron equivalent energy deposition. Note that this energy scale can only be approximated for SuperCDMS HVeV data (see Sec. 2.1.5).

Figure 20: (a, b) (left, large) Energy spectra of excess observations from the individual experiments . In all energy spectra, the rise at low energies is visible. (right, small) Zoom into the excess region of the spectrum [15].

To allow a meaningful comparison of the energy spectra, they were weighted by the energy-dependent cut efficiency of the respective analysis. However, the details of this procedure are different for each measurement: the EDELWEISS, NUCLEUS, SuperCDMS HVeV efficiencies include the trigger efficiency, while the CRESST, DAMIC, Skipper CCD, and SuperCDMS CPD measurements include only the flat survival probabilities above threshold. This leads to apparent peaks in their spectra at 30 eV (CRESST) and 20 eV (SuperCDMS CPD), below which energy the trigger efficiency starts to drop. The concept of trigger efficiency is not applicable to the SENSEI data.

Summarizing the discussion parts in Section 2, the proposed explanations for the excess seem to fall into two main groups. The first one includes sources related to particle interactions in the respective target or surrounding materials, e.g: Cherenkov interactions, luminescence, surface backgrounds, neutrons. Technical or structural issues like stress induced by detector holders, microfractures or intrinsic stress of the target crystals form the second group. Given the strongly varying rates and shapes of the excess signals observed in the presented measurements, a single common origin for them seems to be unlikely. This fact makes it very challenging to pin down the sources of the excess. However, all collaborations are actively testing the above mentioned hypotheses by carrying out dedicated measurements, developing sophisticated veto systems, and improving simulations.

## 4   Summary and Outlook

Achieving extremely low energy thresholds in many rare event search experiments has revealed a yet unexplained excess event rate over known backgrounds, which rises sharply towards the detector thresholds. This led to a common initiative, collecting and comparing data of various measurements among the collaborations joining the EXCESS workshop. In this paper, we summarized 13 individual measurements performed within 10 collaborations, as presented during the workshop in June 2021. We attempt to provide an objective view on the observed data and comprehensively compare the properties of the different measurements. Interpretations and conclusions are left to the readers and will furthermore be the topic of a follow-up event planned for February 2022 [96]. Additionally, a satellite workshop in the course of the Identification of Dark Matter (IDM) 2022 conference is planned. To uncover the origins of the observed excess signals, the community encourages the continuing exchange and discussion of ideas and data, and invites everyone to join the upcoming events planned within this initiative.

## Acknowledgements

The SuperCDMS CPD project has been supported through the National Science Foundation - Grants No. 1809730 and No. 1707704, the U.S. Department of Energy (DOE), Fermilab URA Visiting Scholar Grant No. 15-S-33, NSERC Canada, the Canada First Excellence Research Fund, the Arthur B. McDonald Institute (Canada), Michael M. Garland, the Department of Atomic Energy Government of India (DAE) and the Deutsche Forschungsgemeinschaft (DFG, German Research Foundation)—Project No. 420484612. Fermilab is operated by Fermi Research Alliance, LLC, under Contract No. DE-AC02- 37407CH11359 with the US Department of Energy. Pacific Northwest National Laboratory (PNNL) is operated by Battelle Memorial Institute for the DOE under Contract No. DE-AC05-76RL01830. SLAC is operated under Contract No. DEAC02-76SF00515 with the United States Department of Energy. The SENSEI project is grateful for the support of the Heising-Simons Foundation under Grant No. 79921.

RE also acknowledges support from DoE Grant DE-SC0017938 and Simons Investigator in Physics Award 623940. It was supported by Fermilab under DOE Contract No. DE-AC02-07CH11359. The work of TV and EE is supported by the I-CORE Program of the Planning Budgeting Committee and the Israel Science Foundation (grant No.1937/12). TV is further supported by the European Research Council (ERC) under the EU Horizon 2020 Programme (ERC- CoG-2015 -Proposal n. 682676 LDMThExp), and a grant from The Ambrose Monell Foundation, given by the Institute for Advanced Study. The work of SU is supported in part by the Zuckerman STEM Leadership Program. IB is grateful for the support of the Alexander Zaks Scholarship, The Buchmann Scholarship, and the Azrieli Foundation. TTY is supported in part by NSF CAREER grant PHY-1944826. DB was supported through the U.S. Department of Energy, Office of Science, National Quantum Information Science Research Centers, Quantum Science Center. The MINER work was supported by the DOE Grant Nos DE-SC0018981 for multiple detector R&D efforts for MINER and SuperCDMS, DE-SC0017859 for the Hybrid detector development for MINER, DE-SC0020097 for the cryogenic inner active veto development, and DE-SC0021051 for the partial support to procure the Bluefors fridge. We acknowledge the yearly operations funding provided by the Mitchell Institute. We would like to acknowledge the support of DAE-India through the project Research in Basic Sciences - Dark Matter and SERB-DST-India through the J. C. Bose Fellowship. The EDELWEISS project is supported in part by the French Agence Nationale pour la Recherche (ANR) and the LabEx Lyon Institute of Origins (ANR-10-LABX-0066) of the Université de Lyon within the program "Investissements d'Avenir" (ANR-11-IDEX-00007), by the P2IO LabEx (ANR-10-LABX-0038) in the framework "Investissements d'Avenir" (ANR-11-IDEX-0003-01), and the Russian Foundation for Basic Research (grant No. 18-02-00159). It has received funding from the European Union's Horizon 2020 research and innovation programme under the Marie Skłodowska-Curie Grant Agreement No. 838537. The NUCLEUS experiment is funded by the CEA, the INFN, the ÖAW and partially supported by the MPI für Physik, by the DFG through the SFB1258 and the Excellence Cluster ORIGINS, and by the European Commission through the ERC-StG2018-804228 "NU-CLEUS". The RICOCHET project has received funding from the European Research Council (ERC) under the European Union's Horizon 2020 research and innovation program under Grant Agreement ERC-StG-CENNS 803079, the French National Research Agency (ANR) within the project ANR-20-CE31-0006, the LabEx Lyon Institute of Origins (ANR-10-LABX-0066) of the Université de Lyon. A portion of the work carried out at MIT was supported by DOE QuantISED award DE-SC0020181 and the Heising-Simons Foundation. A portion of this work carried out at Northwestern University was supported in part under NSF Grant PHY-2013203. This work is also partly supported by the Ministry of science and higher education of the Russian Federation (the contract No. 075-15-2020-778). The NEWS-G experiment acknowledges the help of the technical staff of the Laboratoire Souterrain de Modane. This work was undertaken, in part, thanks to funding from the Canada Research Chairs program, as well as from the French National Research Agency (No. ANR15-CE31-0008).This project has received support from the European Union's Horizon 2020 research and innovation programme under grant agreements No. 841261 (DarkSphere) and No. 895168 (neutronSPHERE), and from the UK Research and Innovation - Science and Technology Facilities Council through grants No. ST/S000860/1, ST/V006339/1, and ST/W005611/1. This work has received support from the Arthur B. McDonald Canadian Astroparticle Physics Research Institute. The CRESST project is partly funded by the DFG through the SFB1258 and the Origins Cluster, by the BMBF projects 05A17WO4 and 05A17VTA, by the Austrian Science Fund FWF projects I3299-N36, I5420-N and W1252-N27. FW was supported through the Austrian research promotion agency (FFG), project ML4CPD.

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
