# Peer review of "EXCESS workshop: Descriptions of rising low-energy spectra"

_SciPost Physics, doi:SciPost Phys. Proc. 9, 001 (2022)_

## Round 1 · Referee Report · Anonymous (Referee 1) · 2022-6-10

Strengths

1- This report provides a novel and synergistic link between different experiments by discussing and comparing their observations of an excess of events at low energies. 2- Signals at these low energies (at the 100 eV scale) are indeed a new issue for detectors in this research field to contend with. Sophisticated detectors have been devised, built and operated to detect signal/background events with remarkably low thresholds, in some cases down to the single electron (electron-hole) level. This reports collects and summarizes the relevant aspects of each of these experiments.

Weaknesses

1- Because it is a collection of details and measured results from many different experiments, a minor weakness of the report is its somewhat varied styles in different portions of the manuscript. This is, of course, quite understandable. 2- The synthesis of all the measurements (their excess of events) and the comparison of the spectra was shorter than I expected. I guess the conclusions from the EXCESS Workshop are easy to summarize because at this stage they can be condensed as: a) the excess events are yet unexplained; b) there is no single common origin. The proposed explanations fall into two categories: 1) some interaction (physics); or 2) some detector "instrumental" effect. Therefore, this isn't really a weakness of the report. It's just the nature that, at this stage, this is the status.

Report

This report provides a summary of the EXCESS Workshop. I believe the acceptance criteria of this journal have been met. It's a useful scientific contribution because of the strengths noted above.

Requested changes

Being that this is an atypical manuscript, I do not have changes to suggest. I believe it is preferable that this manuscript serves as a snapshot of the status of excess events as reported by each of the experiments that have taken data at these low energies.

  • validity: high
  • significance: good
  • originality: good
  • clarity: good
  • formatting: excellent
  • grammar: excellent

Author:  Felix Wagner  on 2022-06-23  [id 2604]

(in reply to Report 1 on 2022-06-10)

Dear reviewer,

Thank you for your feedback! We appreciate the work, given that the manuscript is quite long and deals with many different detector concepts. Considering your reported strengths and weaknesses, we agree with your representation of them, and your recommendation.

Kind regards,
the editors of the publication

---

## Editorial Decision

published